# Distributed acoustic sensing as a tool for subsurface mapping and seismic event monitoring: a proof-of-concept

Nicola Piana Agostinetti[1,2], Alberto Villa[1], and Gilberto Saccorotti[3]

[1]DISAT, Universitá di Milano Bicocca, Piazza della Scienza 4, 20126, Milano, Italy
[2]Department of Geology, University of Vienna, Vienna, Austria
[3]Istituto Nazionale di Geofisica e Vulcanologia. Via Cesare Battisti 53, 56125 Pisa, Italy

**Correspondence:** Nicola Piana Agostinetti (nicola.pianaagostinetti@unimib.it)

**Abstract.** We use PoroTomo experimental data to compare the performance of Distributed Acoustic Sensing (DAS) and geophone observations in retrieving data to execute standard subsurface mapping and seismic monitoring activities. The PoroTomo experiment consists of two "seismic systems": (a) a 8.6 km long optical fibre cable deployed across the Brady geothermal field and covering an area of 1.5 x 0.5 km with 100m long segments, and (b) a co-located array of 238 geophones with an average spacing of 60m. The PoroTomo experiment recorded continuous seismic data between March 10th and March 25th 2016. During such period, a $M_L$ 4.3 regional event occurred in the southeast, about 150 km away from the geothermal field, together with several microseismic local events related to the geothermal activity. The seismic waves generated from such seismic events have been used as input data in this study, to tackle similarities and differences between DAS and geophone recordings of such wavefronts.

To asses the quality of data for subsurface mapping tasks, we measure the propagation of the P-wave generated by the regional event across the geothermal field in both seismic systems in term of relative time-delays, for a number of configurations and segments. Additionally, we analyse and compare the amplitude and the signal-to-noise ratio (SNR) of the P-wave in the two systems at high resolution. For testing the potential of DAS data in seismic event location, we first perform an analysis of the geophone data to retrieve a reference location of a microseismic event, based on *expert opinion*. Then, we a adopt different workflow for the automatic location of the same microseismic event using DAS data. To asses the quality of the data for tasks related to monitoring distant events, we retrieve both the propagation direction and apparent velocity of the wavefield generated by the $M_L$ 4.3 regional event, using a standard plane-wave-fitting approach applied to DAS data.

Our results indicate that: (1) at a local scale, the seismic P-waves propagation (i.e time-delays) and their characteristics (i.e. SNR and amplitude) along a single cable segment are robustly consistent with recordings from co-located geophones (delay-times $\delta t \sim 0.3$ over 400 m for both seismic systems); (2) the DAS and nodal arrays are in mutual agreement when it comes to site amplifications, but that it is not immediately clear which geological features are responsible for these amplifications. DAS could therefore hold potential for detailed mapping of shallow subsurface heterogeneities, but with the currently available information of the Brady Hot Springs subsurface geology, this potential cannot be quantitatively verified (3) the interpretation of seismic wave propagation across multiple separated segments is less clear, due to the heavy contamination of scattering sources and local velocity heterogeneities; nonetheless, results from the plane-wave fitting still indicate the possibility for a

consistent detection and location of the distant event; (4) automatic monitoring of microseismicity can be performed with DAS recordings with results comparable to manual analysis of geophone recordings in case of event within or close to the DAS system (i.e. maximum horizontal error on event location around 70 m for both geophones and DAS data) ; and (5) DAS data pre-conditioning (e.g., temporal sub-sampling and channel-stacking) and dedicated processing techniques are strictly necessary for making seismic monitoring procedures feasible and trustable.

## 1  Introduction

Distributed acoustic sensing (DAS) is a novel technology that records ground vibrations, necessary for seismological investigations of the shallow crust (You, 2010; Jousset et al., 2018). DAS technology can be used to record vibrations of a optical fibre cable in terms of the temporal derivative of the strain along the cable direction (Wang et al., 2018). DAS recordings of ground motion along a fibre cable can be obtained at unprecedented high-resolution (less than one meter, if necessary) without the need of deploying tons of instruments on the ground (Henninges et al., 2021). Moreover, DAS recordings can be obtained for Earth's regions which could be unfeasible to reach with standard seismological instruments (e.g. ocean floor, Lindsey et al., 2019). High-resolution seismic recordings are necessary for both the characterisation of next-gen geo-resources (e.g. Enhanced Geothermal System, EGS, or reservoirs for CO2 storage), and the monitoring of geo-resources exploitation (e.g. monitoring of induced seismicity in geothermal wells, Li and Zhan, 2018).

The potential of DAS recordings to be used as "standard" seismic signals has been investigated in previous works (e.g. Yu et al., 2019). In general, DAS recordings can be used for locating off-shore seismic events (Sladen et al., 2019) and reconstructing geological structures in the shallow crust (e.g. Ajo-Franklin et al., 2019). However, some concerns have been raised. For example, van den Ende and Ampuero (2021) highlights the difficulties in using DAS recordings with beam-forming techniques for locating events at large epicentral ranges (i.e., using the fibre as a standard seismic array). Moreover, partial coupling of the cable to the ground, especially for dark fiber, could be a challenge in harsh evironments (e.g. Sladen et al., 2019).

The PoroTomo experiment investigated the Brady geothermal field with both conventional seismic instruments (238 short period 3-components seismometers) and 8.6 km of optical fibre cable (Feigl and The PoroTomo Team, 2017, see also: http://geoscience.wisc.edu/geoscience/people/faculty/feigl/porotomo/). PoroTomo seismic data and DAS recordings have been previously analysed in numerous other studies (Zeng et al., 2017; Parker et al., 2018; Wang et al., 2018; Jreij et al., 2021) In particular, Wang et al. (2018) strictly compares the signal recorded from DAS channels and co-located seismometers for a $M_L$ 4.3 regional event occurred near Hawthorne (about 150 km South-South-East from the test-site). The authors tested different theoretical approaches, like matching the difference of two geophone waveforms with the amplitudes of the sum of the DAS channels in-between, obtaining promising results. From application of template matching to DAS recordings, Li and Zhan (2018) detected microseismic activity at SNR below 1, which was then compared to the pumping history of the geothermal field (Cardiff et al., 2018).

In this study, we move a step-forward and test the potential of DAS recordings to produce data for specific tasks, namely: "subsurface mapping" and "detection and location of seismicity". For "subsurface mapping", we mean the definition of spatial

variations of elastic properties in the sub-surface at high-resolution. With "seismic event detection and location", we have in mind those activities adopted in real-time monitoring procedures. We build on the previous studies and we assume that DAS recordings are coherent to standard seismographic recordings, in the sense that the fibre cable is able to sense elastic waves traversing the rocks where the cable is deployed. From this starting point, we investigate: (1) how standard observables used in subsurface mapping (and their spatial variation across the geothermal field) can be extracted from DAS recordings; and (2) how DAS recordings perform in the automatic detection and location of earthquakes at two different scales of epicentral distance, local and regional. In both cases, we strictly compare the observations obtained using DAS recordings, with similar analysis performed on co-located seismometers. We anticipate here that the main differences of the results obtained with DAS recordings, with respect to standard seismology, are: (1) the necessity for treating a huge amount of data; and (2) the availability of horizontal recordings only, which requires particular care by the time of interpreting results obtained from standard seismic analysis tools.

## 1.1   Geological background and the PoroTomo experiment

The Brady geothermal system is located in the Hot Springs Mountains, northwestern Nevada, USA. The geothermal reservoir lies within layered Tertiary volcanic rocks covered by a sedimentary layer of variable thickness (Jolie et al., 2015). The bottom of the geothermal reservoir is constituted by a Mesozoic metamorphic and plutonic crystalline basement, which is found at depths between 600 and 1500 m below the surface.

The main structural feature of the area is the Brady's fault zone, a fault system extending for some 10km with a dominant NNE strike and steeply dipping to the NW. The geothermal system is strictly related to the presence of those steeply dipping normal faults, where dilatation at depth promotes the fluids circulation (Faulds et al., 2006). The Brady geothermal system is the only one in the area that displays surface geothermal phenomena, for about 4–km along the Brady fault zone. In the area of interest, such geothermal phenomena (e.g. fumaroles warm-grounds and mud-volcanoes) are found, mainly distributed in-between the surface expression of two branches of the Brady fault (Figure 1). The Brady geothermal field is classified as *high entalpy* geothermal resource, reaching a temperature close to $180°$ (Shevenell and De Rocher, 2005).

PoroTomo experiment (Feigl and Parker, 2019) covers the Northwestern part of the Brady geothermal field where subsidence have been observed in the past (Ali et al., 2016). In the PoroTomo area, close to the locations of several injection wells, water circulation models suggest that almost horizontal water flow at shallow depth should occur from the North-West toward the southern portion of the geothermal field. The PoroTomo experiment tested different technologies and different tools for the analysis of the ground deformation of the Brady geothermal field, during a predefined sequence of human activities at the geothermal wells (e.g. repeated injections). All geophysical data recorded and collected during PoroTomo experiment can be found on GDR repository (Feigl and POROTOMO team, 2016), together with information on the geothermal operations. Among the others, data from the PoroTomo experiment has been used for: comparison of catalogs of microseismicity and pumping records on a daily basis (Cardiff et al., 2018), detection of microseismic events through template matching applied to DAS data (Li and Zhan, 2018), active seismic tomography (Parker et al., 2018), comparison of DAS and geophone data for

regional seismic waves (Wang et al., 2018) and evaluation of beamforming potential in DAS data (van den Ende and Ampuero, 2021).

## 2 Data and Methods

We make use of DAS recordings acquired during March 2016 in the framework of the PoroTomo experiment (Feigl and The PoroTomo Team, 2017). The optical fibre cable was deployed with a zig-zag geometry composed of 71 contiguous segments; on average, the length of individual segments was on the order of 150 m. Relevant segments used in the next sections are numbered in Figure 1. Acquisition gauge length is fixed to 10 m, with channel sampling every 1 m, (Wang et al., 2018).

### 2.1 Subsurface mapping

For the subsurface mapping tasks, described in section 3.1, we use one channel every gauge length (10 m) and discard the channels too close (10 m) to the bending point of the cable (as done in Wang et al., 2018). Given the segment length and the gauge length, we obtain about 10-15 independent acoustic records for each segment. The acoustic signal are recorded at 1000 sps. DAS data are organised in 30-seconds long HDF5 files, about one 1Gb for each file. In parallel, we also downloaded and analysed data recorded by a co-located array of 238 Nodal geophones (Fairfield Nodal ZLand 3-component short period seismometers, with a peak frequency of 4.5Hz, called "nodes" hereinafter). Nodes had been deployed as close as possible to the fibre, allowing a detailed comparison of the signal recorded by the DAS and nodes systems.

We make use of the waveforms relative to the $M_L$ 4.3 Hawthorne event. In this case, the analysis is limited to P-wave arrival. P-wave can be clearly seen in both DAS ad Nodes recording (Figure 2). Here, we filter the waveforms between 0.5 and 2 Hz, enhancing the earthquake signal. DAS recordings are downsampled to 500 sps, to be consistent with nodes recordings. To compare nodes to DAS recordings, horizontal components of the nodes are rotated to match the azimuth of the closest segment of fibre cable (Figure 1). All computations on nodal recording are operated on the rotated seismograms, if not specified.

The analysis of the $M_L$ 4.3 event consists in the automatic determination of the relative P-wave arrival times for each segment of fibre cable and for all co-located nodes. The P-wave time delays are computed following the approach described in VanDecar and Crosson (1990) and in Piana Agostinetti and Martini (2019). For each of the three experiments (see Section 3.1), we compute the P-wave time delays for the selected DAS channels (one every 10 m). Following VanDecar and Crosson (1990), we compute: (1) the time-delay between all couples of selected channels, cross-correlating a 2 s long time window; (2) the absolute time-delay $\delta t_i$ for the $i$-th selected channel under the condition of $\sum_i \delta t_i = 0$; and finally (3) the uncertainty on the absolute time-delay $\delta t_i$ following the Bayesian approach described in Piana Agostinetti and Martini (2019), where, for each couple of channels, a family of relative P-wave time-delays are retrieved using a Markov chain Monte Carlo sampling, and posterior estimates of the standard deviations are obtained from such family. The same procedure is performed for the oriented horizontal recordings of the co-located nodes. After the definition of the time-delays, we measure the SNR for each channel and each node, considering a 5 s long time-window before and after the P-wave arrival (Figure 2). On the same time-window

that contains the P-wave, we also measure the maximum amplitude.

## 2.2 Seismic event detection and location

For the detection and location tasks, we consider two test cases of earthquakes recorded at regional and very local ranges. For the former case, we use the aforementioned Hawthorne earthquake. For the latter one, we consider the list of micro-earthquakes obtained by Li and Zhan (2018) from template-matching of 5 events catalogued by the Northern California Earthquake Data Center (UC Berkeley Seismological Laboratory, 2014).

### 2.2.1 Local Earthquakes

DAS recordings of the local earthquakes are band-pass filtered over the 2-20Hz frequency band, and resampled at 100Hz. A spatial sub-sampling is then performed by stacking groups of 11 adjacent channels within each segment, with a 30-channel step. The 10 channels before and after each corner are excluded. Resulting from this procedure is a virtual array whose total number of channels (239) is similar to that of the nodal array. For each stacked trace, we calculate a characteristic function given by the Kurtosis (Langet et al., 2014), and use an AIC autopicker (Sleeman and van Eck, 1999) to automatically identify the onset time of these functions. Finally, we use the DBSCAN algorithm (Ester et al., 1996) to discard the most obvious outliers. The same procedure is adopted for the automatic estimate of P-wave arrival times at the vertical components of the nodal array.

For each event, reference locations are obtained from inversion of manually-picked P- and S-wave arrival times estimated at stations of the nodal array and, when available, from the UC Berkeley Seismological Laboratory (2014) catalog. Hypocentral locations are retrieved using the probabilistic, non-linear approach coded into the NonLinLoc software package (Lomax et al., 2009); the likelihood function for source location is explored using the Octet-tree sampling method.

### 2.2.2 Regional Earthquake

For the Hawthorne earthquake, 2-minute-long DAS recordings are band-pass filtered over the 0.5-2Hz frequency band, and resampled at 100Hz. Spatial sub-sampling is performed in a similar manner to what described above, but stacking over 51 adjacent channels with a 50-channel step, thus obtaining a virtual array of 102 elements. Delay times $\Delta T_{ij}, i \neq j$ between all the independent array channels are derived from the maxima $CC^{max}$ of the corresponding cross-correlation function. These differential times are used to derive the horizontal slowness vector following the Plane Wave Fitting (PWF) method (Del Pezzo and Giudicepietro, 2002):

$$\Delta \boldsymbol{T} = \Delta \mathbf{x} \cdot \mathbf{s} \tag{1}$$

where the matrix $\Delta\mathbf{x}$ contains the differences between the $x$ (EW) and $y$ (NS) components of the array channel's coordinates, and $s$ is the horizontal slowness vector, from which we derive the propagation azimuth (measured clockwise from the N direction) and apparent velocity.

Equation 1 is solved using a weighted least-squares approach, with weights defined as:

$$w_{ij} = \frac{CC_{ij}^{max}}{1 - CC_{i1}^{max}} \tag{2}$$

so to emphasise the contribution of the most correlated channels. For the inversion, we considered only those channel pairs exhibiting a $CC^{max}$ larger than an arbitrary threshold, here set equal to 0.85. The procedure is iterated over 4-s-long time windows, sliding along the DAS recordings with 80% overlap. Slowness uncertainties, and corresponding errors in azimuth and ray parameter estimates, are derived assuming a constant error of twice the sampling interval (i.e., 0.02s).

## 3 Results

### 3.1 Subsurface mapping

Direct comparison of DAS and Nodal recordings is not possible, due to the sensitivity of the two seismic systems to different geo-observables. Many different procedures have been developed to convert DAS recordings to Nodal recordings and viceversa (Wang et al., 2018, and references therein). Here we do not aim to use the two systems together in the same analysis, but to compare the performance of the two systems on the same analysis workflow. Thus we do not transform one system into the other, but we use them separately. However, for only one case, we start our analysis reproducing the results obtained in Wang et al. (2018) for comparing the two systems using the amplitudes of the sum of several DAS strain-rate waveforms along a segment and the finite difference of two geophone waveforms at the two ends of the same segment. We use the DAS data recorded along segment 15, where 4 nodes have been co-deployed (Figure 3). This allows us to divide the segment 15 into three sub-segments and analyse each sub-segment as an independent set of information (i.e. only data from the four geophones have been used twice). We make use of the workflow of Wang et al. (2018, Figure 9 and equation 5), composing a "representative DAS waveform" from the sum of one channel of DAS data every 10 m in between two geophones and comparing it to the "finite-difference Nodal waveform" of the two geophone waveforms. Our results confirm the finding in Wang et al. (2018), where the authors indicate that the amplitude recorded by the two systems are coherently correlated. We also observe, as found in Wang et al. (2018), a small absolute time-delay between the two systems, that we correct cross-correlating the "representative DAS waveform" with the finite-difference Nodal waveform.

The propagation of the P-wave generated by the $M_L$ 4.3 event can be easily tracked across the PoroTomo investigated area using the vertical components of the geophones. Such analysis can be used as a reference of the general propagation direction and time-delay. To get a first insight on the potential local anomalies in the P-wave propagation, we process vertical and North components of the nodes. In this case, we make use of the same tool (i.e. cross-correlation of the waveforms following VanDecar and Crosson, 1990) without rotating the horizontal as described in the previous section. Vertical components of the

nodes clearly display similar P-wave arrivals, and the cross-correlation procedure gives correlation coefficients as high as 0.96-0.99 on average (Figure 4). Mapping the time-delay obtained through with the application of VanDecar and Crosson (1990) s approach shows a South-South-East to North-North-West propagation as expected (Figure 5a). On the contrary, repeating the same analysis for the North component of the nodes and mapping the results, we observe that the wave propagation is more complex showing, for example, an area in the center of the nodal array, where strong negative time-delay are observed (Figure 5b). Those could be apparent anomalies given by local surface waves generated from interaction of the P-wave with the local topography and erroneously cross-correlated with the correct P-wave. Nevertheless, this result supports our workflow where DAS data need to be strictly compared to co-located, re-oriented Nodal data, i.e. nodal horizontal components need to be vector summed in DAS direction.

We first present the analysis on the P-wave recorded by the DAS system, considering DAS segment 48. This segment is the longest one (about 350 m), with 7 geophones almost co-located along the cable (within 30m), and it gives us the possibility of following the P-wave over a long distance (Figure 7). P-waves recorded along the cable are generally similar, but not as much as on the vertical components of the Nodal system, giving smaller cross-correlation coefficients ( about 0.95 on the average, Figure 6), with some waveforms displaying large pre-signal noise (channel 5534 and 5524 in Figure 6a). Mapping the time-delays and the other quantities for the two systems shows that the time-delays are generally consistent, considering the computed uncertainties. Time-delays decrease from 0.19 s to -0.05 s (with a minimum of -0.10 s) for the nodes and from 0.19 s to -0.15 s for the DAS data. In particular, only one geophones (N051) does not follow the uncertainites in time-delay computed from the DAS recordings. However, such geophone could have had issues (see also Appendix 1). Noteworthy, vertical components of the nodal system display more limited decrease in P-wave delay-time going from South to North, between 0.11 and s and -0.04 s, confirming that vertical and horizontal components give different measurements, as found above. Maximum amplitude in the two systems also displays similar spatial trends. More interesting, SNR in the DAS system has a sudden drop at about X=-100 m in our profile projection (Figure 7d), where we also observe the nosiest waveforms (Figure 6a), possibly indicating a partial failure in the cable coupling to the ground. However, such defect (if existent) does not bias the time-delay measurement.

We repeat the same experimental setting for three, near parallel and consecutive, DAS segments: 28, 33, 68. Those segments have almost the same azimuth of N30W, they are aligned but are not contiguous, and cover a 500 m long profile. Nine geophones have been deployed close to such segments (Figure 8). Also in this case, the spatial variations of time-delay, SNR and maximum amplitude between the two systems is coherent. Time-delays on the horizontal components of the nodal system display values between -0.05 s and 0.15s, a larger value with respect to what we could expect from standard estimations (considering a 450 m long cable and an average apparent P-wave speed of 5 km/s (van den Ende and Ampuero, 2021, consistent with beamforming analysis). SNR does not vary significantly, with a small decrease toward South. Maximum amplitude displays a sharp decrease from North to South in both system, in the very first hundred meters. The two systems are not perfectly coherent, but 1 geophone out of 3 is far from the cable (about 50 m). Time-delay shows contrasting evidence in the two system at the end of the profile toward South, even considering the computed uncertainties. While the overall trends in the two system is coherent (i.e. largest values in the central portion of the profile and smallest at the two ends), a 60 m long cable section of segment 68 show time-delays as small as -0.20 s, not found in the geophone data. A close inspection to the waveforms (Figure 8ef) reveals

the presence of possibly scattered waves in DAS recordings at channel 8179 (i.e. strong differences from channel 8199, not found in the Nodal system, where differences in the recordings of co-located nodes, N046 and N069, are minimal).

To test the possibility of appreciating 3D features, we apply our workflow to two parallel segments but not aligned, 3 and 5 that run along the longer side of a rectangular area of about 150 x 70 m. Eight geophones are deployed along the two segments (4 geophones/segments, Figure 9). The results indicate that both time-delays and maximum amplitude display the same spatial variation. In particular, time-delay ranges between -0.12 s and 0.12 s for both systems and indicate an un-expected East-to-West propagation. Amplitude variation are coherent in both system, increasing toward West. Both P-wave delay-times and amplitude spatial variation correlate with the P-wave velocity gradient found in Parker et al. (2018). In such velocity model, North-South oriented velocity anomaly is found at shallow depth (Figure 8ab in Parker et al., 2018), potentially related to the local fault stystem.

In this case, we notice that channels close to the end of the segments are generally different from the closest one. We suggest that the practise of removing the channels closer to the bending point of the fibre should be revised increasing the lag distance, now 10 m (a lag of 15 m or 20 m could work finely).

Finally, we analysed separately all segments between 1 and 27, covering the Northwestern side of the PoroTomo experiment, where most of the geothermal phenomena are found bounded by two faults mapped at the surface (i.e. fumaroles and warm-ground, Figure 10). The variations of the maximum amplitude show that the highest values, i.e where maximum amplitude goes beyond the threshold of 0.1, are found in the only two places where the fibre cable crosses the geothermal active area (i.e. near the northernmost fumaroles and where warm ground is mapped) roughly traversing the two bounding faults. Given the nature of the geothermal phenomena, we suggest that amplitude variations in this case could be related to local site effects associated with the development of the underground fracture network. Comparison with available velocity models show controversial details. On one hand, the area with the highest amplitude perfectly correlates with a very well -defined volume showing very low $V_P$ values with respect to the surrounding rocks (Parker et al., 2018; ant PoroTomo Team, 2018). Conversely, the second interesting area, more to the North, does not show any relevant velocity anomaly. In such area, where all the injection wells are located, we were expecting a well-developed fracture network and, thus, a low velocity rock volume..

## 3.2 Seismic event detection and location

### 3.2.1 Local Earthquake

For comparing the performances of the nodal and DAS deployment toward the automatic location of micro-earthquakes at short distance ranges, we focus on an earthquake reported in the template matching earthquake catalog of Li and Zhan (2018), with detection time 2016 March 14 at 10:42:07UTC. Following a preliminary inspection of arrival times, that event appeared to be located in close proximity of the DAS and nodal deployments, in turn exhibiting good visibility of both waveforms and cross-correlations (see also Supplementary Materials in Li and Zhan, 2018).

For deriving a reference location, precise P- and S-wave arrival times are estimated manually at 75 vertical and 48 horizontal channels of the nodal array. P-wave theoretical travel times are calculated in a 1-D gradient velocity model, obtained after

averaging the 3D velocity structure reported by University of Wisconsin (2015) (see Supplementary Materials, Fig. S1). The corresponding S-wave travel times are derived considering a ratio between P- and S- wave velocities ($V_P/V_S$) equal to 2.6, as indicated by the modified Wadati diagram (Supplementary Material, Fig. S2).

Figure 11 illustrates the comparison between the reference location obtained from manual picking of nodal data (Figure 11a), and those derived from the automatic processing of both the DAS and nodal arrays (Figure 11bc). Location from the P- and S-wave inversion indicates a well-constrained source volume located at the center of the SW side of the deployment, at a depth of 460[±49]m beneath the surface (Figure 11a). Horizontal uncertainties are on the order of 63m and 75m for the EW- and NS-coordinate, respectively; the root-mean-square of residuals (RMS) is 0.036s.

Inversion of P-wave automatic pickings from both the DAS and nodal arrays reports a similar epicentral region. These solutions, however, are clipped at the surface, as a consequence of two separate facts. The first, is that the automatic pickings at both DAS and nodal arrays include slower arrivals from more distant stations, which had been discarded from the manual estimates (see Figure 11d). In addition, the manual location takes advantage of S-wave arrival times, which provide a solid constrain to ray length and hence to source depth.

Additional examples of locations derived from inversion of automatic first-arrival time pickings at the DAS virtual array are reported in the supplementary material, for events either catalogued by NCEDC (2014), or reported in the list of template matching detections from Li and Zhan (2018). In all these experiments, the performances of the DAS data to retrieve the 'correct' location are comparable to what exhibited by the nodal deployment. However, a more comprehensive assessment of these results is made difficult by the fact that all the sources are located well outside of the deployments. Under such condition, and in absence of S-wave time pickings, the ability to retrieve the source-to-receiver distance depends on the gradient of the velocity structure, which controls the geometry of the rays and hence their intersection with the surface.

### 3.2.2 The Hawthorne earthquake

In addition to what described in the previous Section, the DAS recordings from the Hawthorne earthquake have already been analyzed in Wang et al. (2018) and van den Ende and Ampuero (2021). In particular, this latter study compared the performance of the DAS and nodal arrays in seismic beamforming for deriving the propagation parameters (apparent velocity and propagation azimuth) for the direct P- and S-wave arrivals. In their work, van den Ende and Ampuero (2021) applied the MUSIC method (Goldstein and Archuleta, 1987) to different combination of DAS channels; in no case, however, did they obtained results compatible with what expected for a single plane wave propagating along the source-to-receiver direction. This occurrence was attributed to marked velocity heterogeneities and scattering sources, locally distorting the wavefront and producing loss of signal coherence, in turn masking the weaker P-wave signal impinging at the array with steep incidence angles. Following this observation, van den Ende and Ampuero (2021) concluded that '....*the DAS array exhibits poor waveform coherence and consequently produces inadequate beamforming results that are dominated by the signatures of shallow scattered waves*'.

Although these conclusions are fully consistent with the complex wave propagation we outlined above, our application of the PWF approach over short time windows sliding along the sub-sampled DAS recordings yields to promising results for which concerns the real-time detection of sources at regional distance (Figure 12) . As a matter of fact, the onset of the earthquake

signal is marked by an abrupt increase of the average multi channel correlation (Figure 12b) which, in correspondence of the
P-wave arrival, peaks to a value which is about 2 times larger than those associated with the preceding background noise.
The P-wave propagation azimuth associated with the most-correlated arrivals are biased by some $30°$ with respect to what
expected from the source location. The apparent velocities associated with the most-correlated portion of the P-wave arrival
are consistent with what estimated by van den Ende and Ampuero (2021) using data from the nodal array. Similar consistency
is observed for the S-wave arrival, even if its onset is less clear due to contamination by the P-coda wavetrain.

Overall, the results from this exercise suggest that (i) when dealing with seismic wavelengths comparable to the DAS
aperture, a simple thresholding on the overall correlation of DAS channels may serve as an efficient operator for detecting
the arrival of an earthquake signal; (ii) although with some bias, the estimate of the P-wave direction of propagation and
apparent velocity is reliable enough to obtain a preliminary estimate about the back-azimuth to the source, and (iii) a severe
correlation-based weigthing scheme is required to emphasize the delay times retrieved from the most coherent channel pairs.

## 4   Discussions

Our analysis confirms that DAS recordings can be used as complementary or in substitution of standard geophones for both
monitoring activities and exploration of the the subsurface. Taking into account the limitations of DAS recordings, several
challenges should be addressed in the future for enabling the use of such data in, e.g., standard monitoring routines. Those
challenges arise mainly from the intrinsically different structure of the data obtained from the fibre cable with respect to what
is recorded from a set of single-vertical or 3C geophones, more than from the two different physical observables recorded by
DAS and geophones. DAS data structure is characterised by a huge amount of observations of the seismic waves along (1) an
horizontal axis, and (2) with a relatively poor areal coverage with respect to a standard seismic network. Exploiting such data
structure would probably require to re-think the "old style" algorithms used seismic data analysis, which have been developed
since the '70 mostly for areal distributions of individual vertical geophones. New approaches developed for large N deployment
of geophones (e.g. Long Beach) could be more appropriate and easy to adapt to DAS data.

Our analysis of the DAS and geophone data substantiates previous findings that horizontal and vertical recordings gives dif-
ferent images of the propagation of distant events across a seismic array. This is a fundamental point to be kept in mind when
analysing DAS data, which are only horizontal, in most-used, standard configuration of the fibre cable. Local scattering phe-
nomena associated with topographic roughness and / or velocity heterogeneities may negatively interfere with the propagation
of the P-wave, potentially overwhelming the first arrivals and biasing the measure of, e.g., relative time-delays.

Looking into the details of the propagation of the P-wave of a regional event along segments of fibre cable, we observe
that local variations of standard measures (time-delays, SNR and maximum amplitude of the signal) are consistent between
fibre cable and co-located geophones. In particular, we found that the analysis of the data recorded by a single segment of
fibre cable shows the most stable results, even in case of poor coupling of the fibre. Analysing the data for multiple, parallel,
not-consecutive segments provides unclear results, likely as a consequence of scattering and wavefront distortion induced by
local velocity heterogeneities. In a case, the two parallel not-aligned segments, results are definitely promising, with potential

for 3D subsurface reconstructions. In the case of a profile composed of three separated segments, results are ambiguous and need further investigations. Analysing each single segments on the North-West side of the PoroTOMO experiment where the geothermal phenomena are found (warm grounds and fumaroles), we show a potential correlation, at a higher resolution with respect to geophones, between amplitude variations of the recorded signal and local near surface geology. However, such correlation should be more quantitatively investigated with integrating a larger amount of geological data, not available to us at the moment, and needs further investigations to be confirmed.

In our experiments for the automatic location of local earthquakes, a basic auto-picking procedure produces similar results once applied to the nodal and DAS data. In general, those time estimates require a preliminary selection before being used for locating seismic events. Moreover, in DAS data seismic phase recognition can be more difficult than in geophone data, due to the absence of vertical recordings which are most sensitive to the P-wave arrival. Converted / scattered waves from the P-wave coda can be easily mis-picked as P-wave at more distant or less favourably-oriented segments, where the background noise can mask the very first onset of longitudinal waves impinging at steep incidence angles. This is clearly observed in Figure 11d: although the DAS pickings are generally consistent with the nodal ones, at short epicentral ranges the former exhibit a larger scatter of the residuals, likely deriving from mis-detections of P-coda waves or even S-waves.

DAS data analysis must thus relies on specific methodologies where the consistence of the data themselves with the under-lying hypothesis is statistically evaluated and accounted for. Within this context, it is worth mentioning a statistical approach developed recently, which consistently defines, in a Bayesian sense, the "membership" of each datum to the "outliers" class, and which could help further developments of DAS data treatment (Tilmann et al., 2020).

In case of wavelengths comparable to the size of the deployment, PWF results from the regional earthquake suggest that reliable event detection and location may be achieved. To that purpose, we deem that the most crucial aspect relies on the severe correlation-based channel selection and weighting scheme adopted in the inversion of differential times. While considering DAS potentialities for wavefield decomposition, we also note that the huge amount of available channels in DAS systems is perfectly suited for application of processing techniques that improve both accuracy and precision of existing frequency-wavenumber methods, such as sub-array spatial averaging (e.g. Goldstein and Archuleta, 1991).

Finally, we voluntarily skip an important task in seismic monitoring systems: the definition of the event magnitude. We are aware that such task would require additional work to be fulfilled; as previously shown, DAS recordings may exhibit large amplitude variations even at close-by channels, thus limiting the potential of using such kind of data for magnitude estimations. Nonetheless, recent studies (e.g. Lior et al., 2021) report promising indications toward utilisation of DAS data for source studies and Magnitude evaluation.

## 5 Conclusions

We make use of DAS recordings obtained during PoroTomo experiment (Brady geothermal field), for testing their potential in simple subsurface mapping and monitoring tasks. We analyse waveforms from a regional events, to get insights into the local

structure of the geothermal field, and from a local microseismic event, to perform precise automatic event location. Our main findings are:

1. DAS recordings can be used for monitoring and subsurface mapping purposes and their performance is comparable to seismological records. For the detection and location tasks, we confirm that pre-processing procedures, such as channels selection based on ground coupling, removal of channels close to cable corners, P- and S- phase identification, band-pass filtering tailored to the wavelength of interest, could be necessary to keep in the process data that satisfy quality requirements.

2. Additionally to quality requirements, data selection is definitely necessary due to the large amount of data in DAS recordings. Taken together with the above point, this fact opens additional questions on the criteria to be followed for data selection and/or spatial/temporal subsampling, a sensitive point for developing automatic data-processing procedures;

3. DAS recordings represent horizontal ground-displacement only, which limits, in some sense, the use of standard seismological analysis tools depicted specifically for the processing of vertical or 3D ground-motion. Within this context, a clear example is offered by the autopicking performances discussed in Section 3.2.1 and Appendix A1-A2.

4. Although advantages and limitations of DAS systems as a seismic antenna need further investigation (van den Ende and Ampuero, 2021), our results indicate a clear potential of DAS toward coherence-based detection of sources at regional distance, complementing the results in Li and Zhan (2018) concerning detection of local microseismicity at low SNR. If used as a seismic antenna, DAS should be deployed according to multiple segments oriented along a large variety of azimuths, so to increase the overall sensitivity toward distinct wavetypes propagating along different directions;

New monitoring and subsurface mapping tools developed for DAS recordings are needed. In particular, exploration of the DAS data-space (Tarantola, 2005; Piana Agostinetti and Sgattoni, 2021; Piana Agostinetti et al., 2021) would give an important contribution toward the analysis of DAS recording in a semi-automatic system.

## Appendix A:  P-wave time-delays from local event

Here, we present the analysis of P-wave delay-times for a local event. This analysis is complementary to the results presented in Section 3.1 because the frequency content of the P-wave of a local event is generally comprised between 10 and 20 Hz. However, due to the close proximity of the event to the stations, we can not consider a plane wave approximation and we have to compute P-wave delay-times using a different approach with respect to the analysis reported in Section 3.1. In Figure A1, we show the geometry of the event, nodal stations and DAS channels along segment 48. We make use of an event and cable geometry to try to maximise the P-wave delay-times along the cable, i.e. the event occurs almost along the segment strike. Recordings of the local events for three systems are considered: vertical nodal stations, oriented horizontal nodal stations and DAS channels. The three systems have been analysed with the same workflow, which encompasses both (1) automatic picking of a P-wave based on classical auto-picker (Baer and Kradolfer, 1987) and (2) manual revision of the automatic pickings. The Baer picker has been developed as a ObsPy package and we set with its default values (Beyreuther et al., 2010).

P-wave on the vertical component of the nodal stations display an high SNR and can be easily detected by the automatic picker (violet vertical bars in Figure A2a, not really visible because below the green bars). One station (N051) has a more limited SNR and identifying P-wave arrival can be more challenging. Manual revision does not sensibly change the arrival times (green vertical bars in Figure A2a). Conversely, the P-wave on the oriented horizontal components of the nodal stations display a smaller SNR, at least for some stations (violet vertical bars in Figure A2b), which need substantial manual revision, at least for N051. Autopicker results and manual revision on DAS channels are more interesting. The P-wave arrival times obtained with the Baer picker (violet vertical bars in Figure A2c) display a coherent pattern from channel 5579 to 5419, the closest half of the segment to the epicenter. In the second half of the segment, the autopicker gives more scattered results. Comparing with the manual pickings (green vertical bars in Figure A2c), which also have a larger variability in the second half with respect to the first half, such automatic pickings seem, in many cases, to arise from the mis-detection of either a receiver-side converted wave, Ps orSp, or the S-wave itself, as a P-wave.

Manual pickings in the three systems are presented in Figure A2d). Manual pickings for both horizontal and vertical components of the nodal stations (grey symbols) are generally similar, with a slightly higher delay-times for the horizontal components. Notably, those delay-times coincide with the delay-times measured along the cable approximately in the same locations (some nodal station have been deployed not exactly on the fiber cable). Delay-times measured along with the DAS show a more complex pattern with coherently higher values in some section of the cable (e.g. between channels 5389 and 5359). This analysis confirms the potential for the DAS recordings in reaching a higher resolution especially if gauge length can be reduced to 1 m , which is already feasible with the next generation of fiber interrogators. However, also in this case, picking procedures, established for a network of vertical components, do not perform with profit, when applied to horizontal recordings.

## Appendix B: Amplitude variations with frequency: the case-study of Vibroseis recordings

To obtain more insights into the amplitude of the signal recorded along the fiber cable, we analyse the signal generated by a Vibroseis truck in a position along the strike of the segment #48. The Vibroseis sweeps T49 considered here are three sweeps generated in the same location with the same apparatus (P-source). Thus, for each geophone and each DAS chanel, we obtain an averaged value for the maximum amplitude of the filtered signals and a standard deviation. The Vibroseis sweeps consists in a 20 s signal of increasing frequency, from 5 Hz to 80 Hz. Here we selected three frequency bands (5-10 Hz, 10-20 Hz, and 20-40 Hx) and we measured the maximum amplitude of the filtered signal as it varies along the cable. In this way, we can highlight local amplification for specific frequency bands.

In Figure A3a), we show the geometry of the source and receivers. Source is indicated as T49. One station (N026) is clearly closer to the source with respect to the cable, and results from this station have been removed. Visual inspection of the three frequency bands (e.g. for one station and one co-located channel, Figure A3b), show that variations observed in the geophone recordings are also mapped in the DAS recordings. The patterns in maximum amplitude for the two systems (oriented horizontal geophones and DAS) as a function of the distance from T49 are shown in Figure A3c. Such patterns are coherent in all the three frequency bands. In particular, for the 10-20 Hz band, local amplifications found in the geophone data is consistently retrieved

from the DAS data (maximum amplitude found at about x=210m). Moreover, DAS data offer high-resolution details clearly

not available from the geophone data (due to their large inter-station distance). Our analysis confirms that DAS recordings can

be used to obtain trustable information on the amplitude of the recorded signal. Again, the high spatial sampling of the fiber

cable allows to extract information un-available using sparse stations.

*Author contributions.* NPA and AV equally contributed to the data analysis in the subsurface mapping analysis. GS developed and executed the monitoring analysis. NPA and GS wrote the original draft of the manuscript.

*Competing interests.* No competing interest exists on the presented research

*Acknowledgements.* We kindly thank Martijn van den Ende, Herbert Wang and an anonymous reviewer for their constructive discussion about DAS data analysis and interpretation. NPA thanks PoroTomo TEAM (Kurt Feigl, Nicole Taverna and Jon Weers) for their continuous support in handling the data. NPA's publications are funded by the Austrian Science Fund (FWF) under Grant M2218-N29. Some figures are plotted using GMT (Wessel and Smith, 1998). PoroTomo data are available at the following weblink: https://dx.doi.org/10.15121/1368198

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

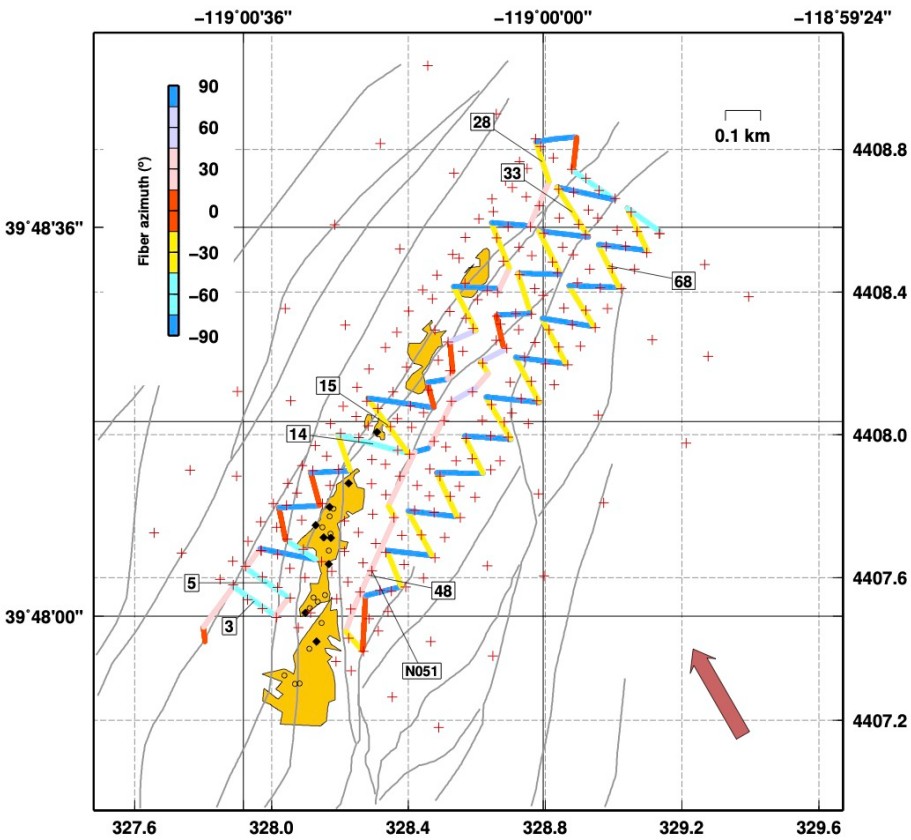

**Figure 1.** Porotomo experiment map: Fibre cable used for DAS (coloured line, colours indicate average azimuth of the segments), Numbers indicate segments described in the text; Nodal stations; faults (grey lines), warm ground (yellow areas); fumaroles (black diamonds) and mud-volcanoes (open diamonds). A red arrow indicates the direction of incoming P-Wave for the Mw 4.3 Hawthorne earthquake.

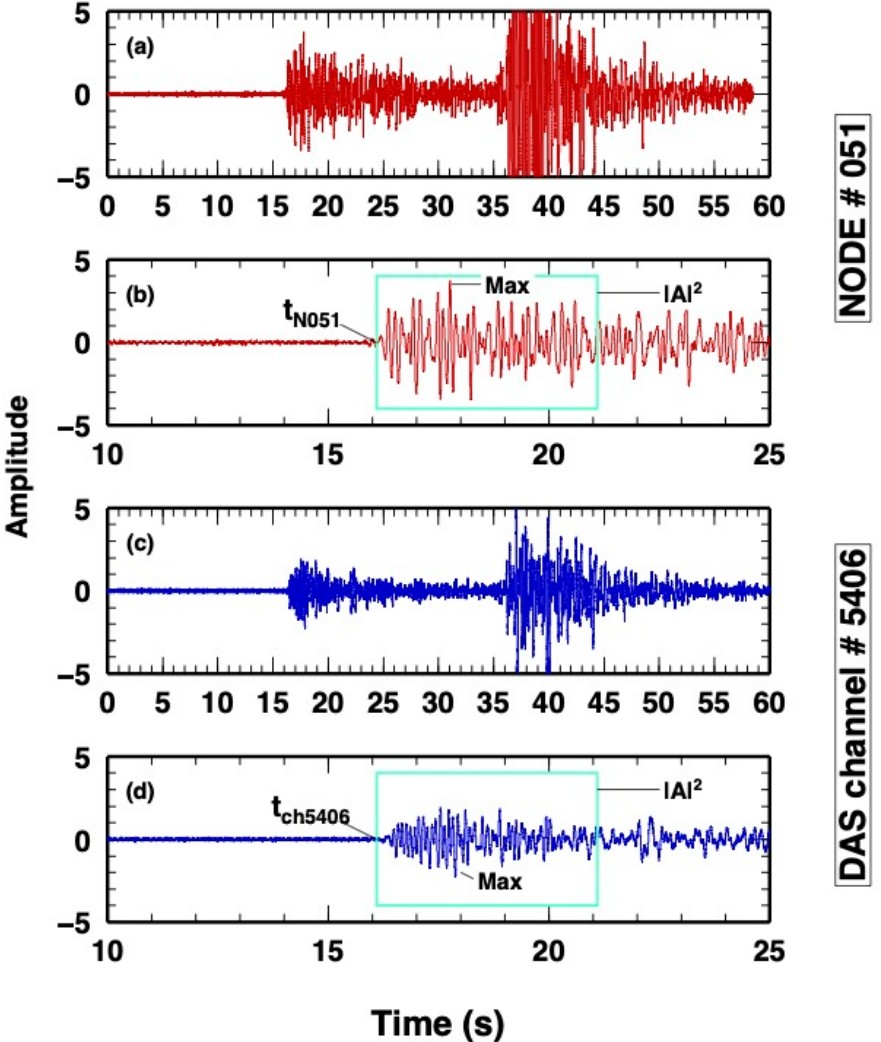

**Figure 2.** Waveforms generated from the Mw4.3 Hawthorne earthquake, with a zoom on P-wave with the quantities analysed in the study: P-wave arrival time, MAX amp, Energy in the P-window (for computing SNR). Node 51: (a) and (b); Closest DAS channel to Node 51 (channel number 5406): (c) and (d)

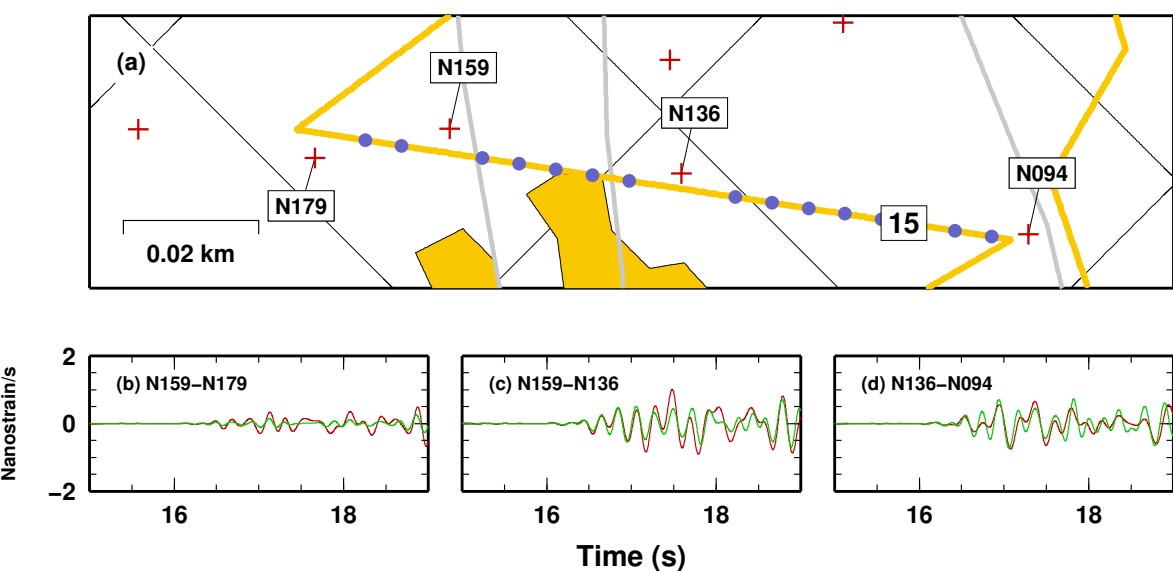

**Figure 3.** Comparison of DAS and NODAL recordings, following Wang et al. (2018) Equation 5. (a) Map of DAS segment 15 and co-located Nodes. P-wave arrival in the three sub-segments as temporal derivative of strain, between: (b) Node 94 and Node 136; (c) Node 136 and Node 159; (d) Node 159 and Node 179.

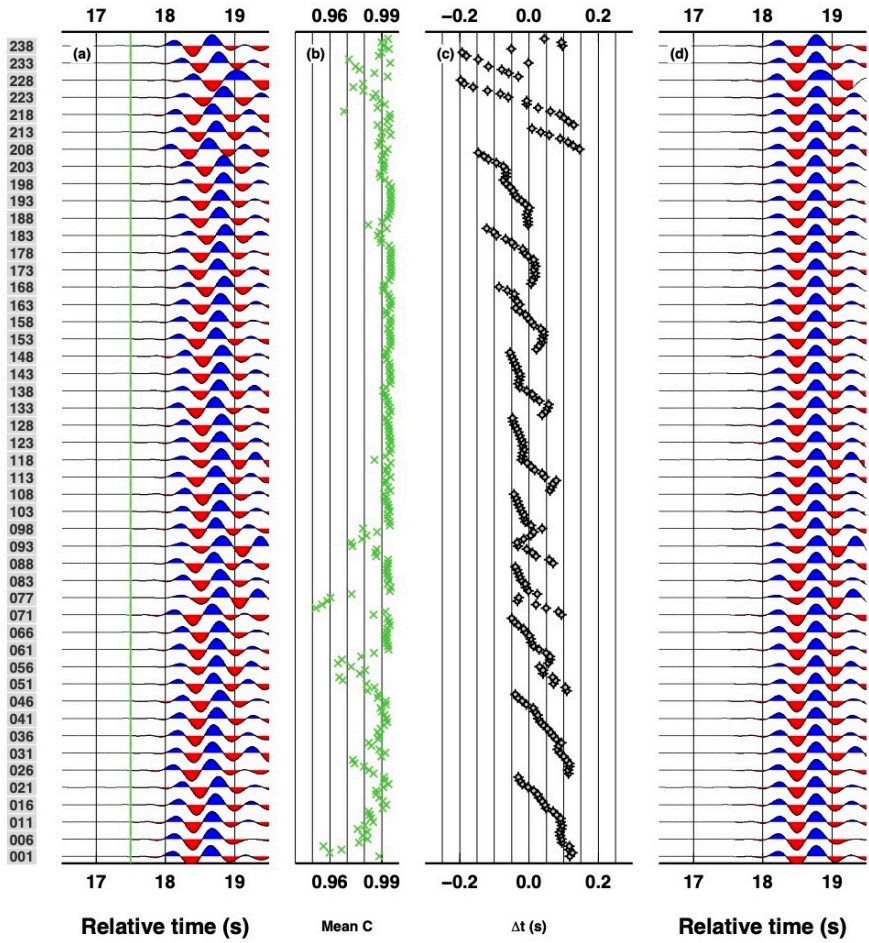

**Figure 4.** Example of analysis for computing P-wave time-delays using VanDecar and Crosson (1990)'s approach applied to Vertical recordings of Nodal seismometers. (a) Original waveforms filtered between 0.5-2 Hz showing P-wave arrivals. (b) Average cross-correlation value, for each station cross-correlated to all others. (c) Relative P-wave time-delays. (d) Aligned P-wave arrival using time-delays in (c). Stations are ordered with station names. Only one every five stations is shown in panels (a) and (d) for clarity.

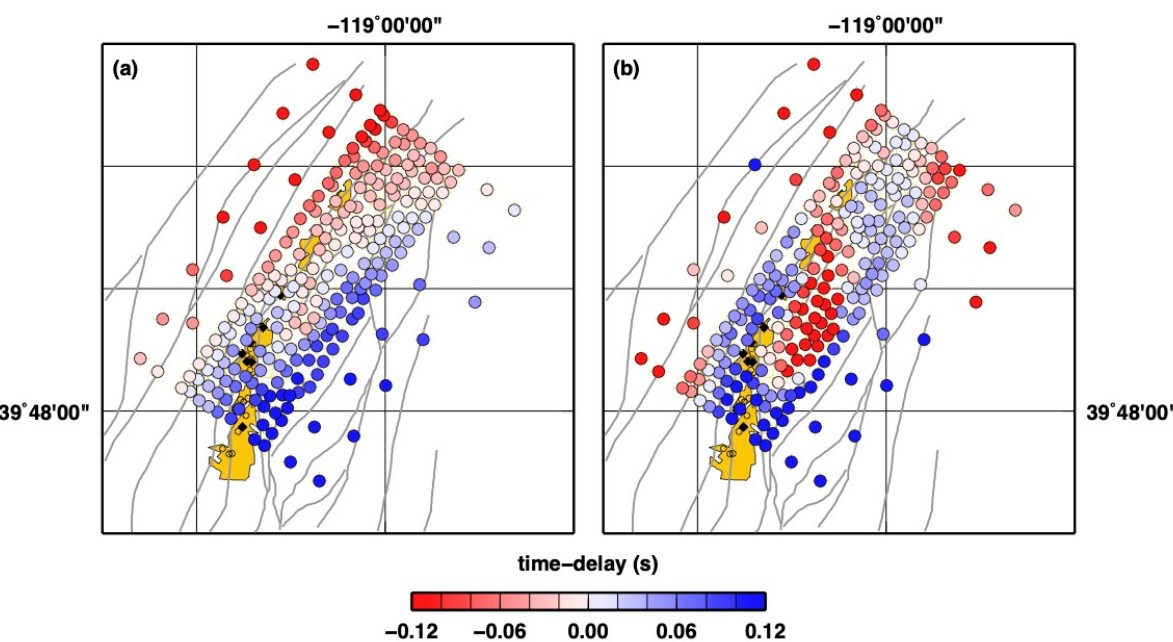

**Figure 5.** Maps of P-wave time-delay from correlation of recordings by Nodal stations. Waveforms have been filtered to 0.5-2.0Hz: (a) Vertical components; (b) North components.

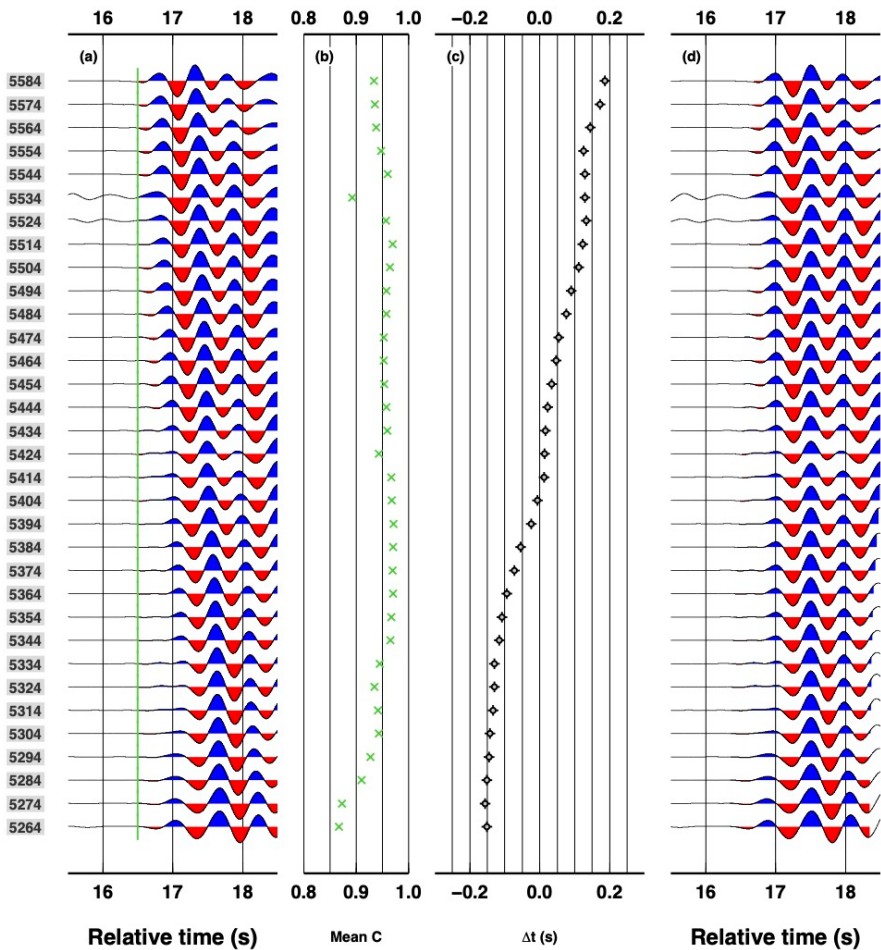

**Figure 6.** Example of analysis for computing P-wave time-delays using VanDecar and Crosson (1990)'s approach applied to DAS channels along segment 48. Same panels as in Figure 4.

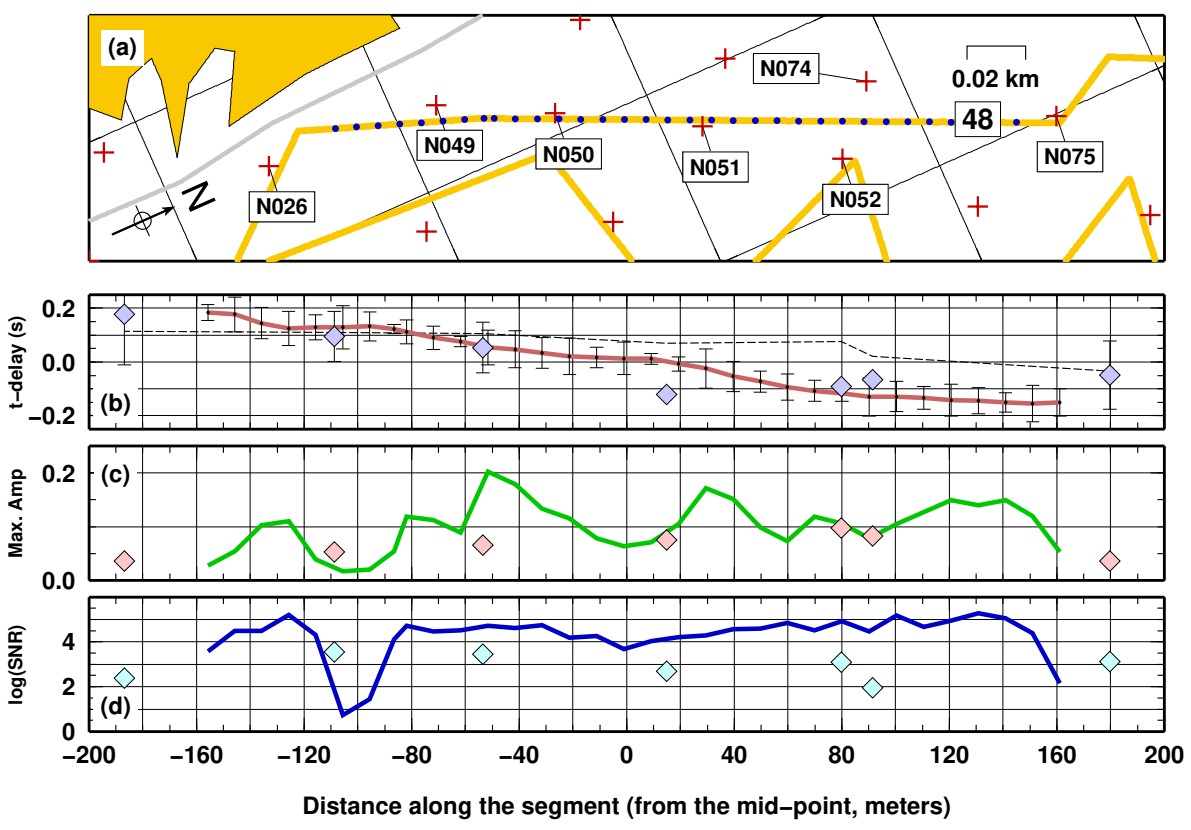

**Figure 7.** Measurements along the single DAS segment 48 (coloured lines), compared to measurements obtained using rotated horizontal recordings of co-located Nodal stations (grey diamonds). (a) Map of the DAS segments. (b) P-wave time-delays with associated uncertainties, together with the P-wave time-delays measured on the vertical components of the nodes (thin dashed black line); (c) maximum amplitude; and (d) log(SNR) as a function of the distance along the cable segment.

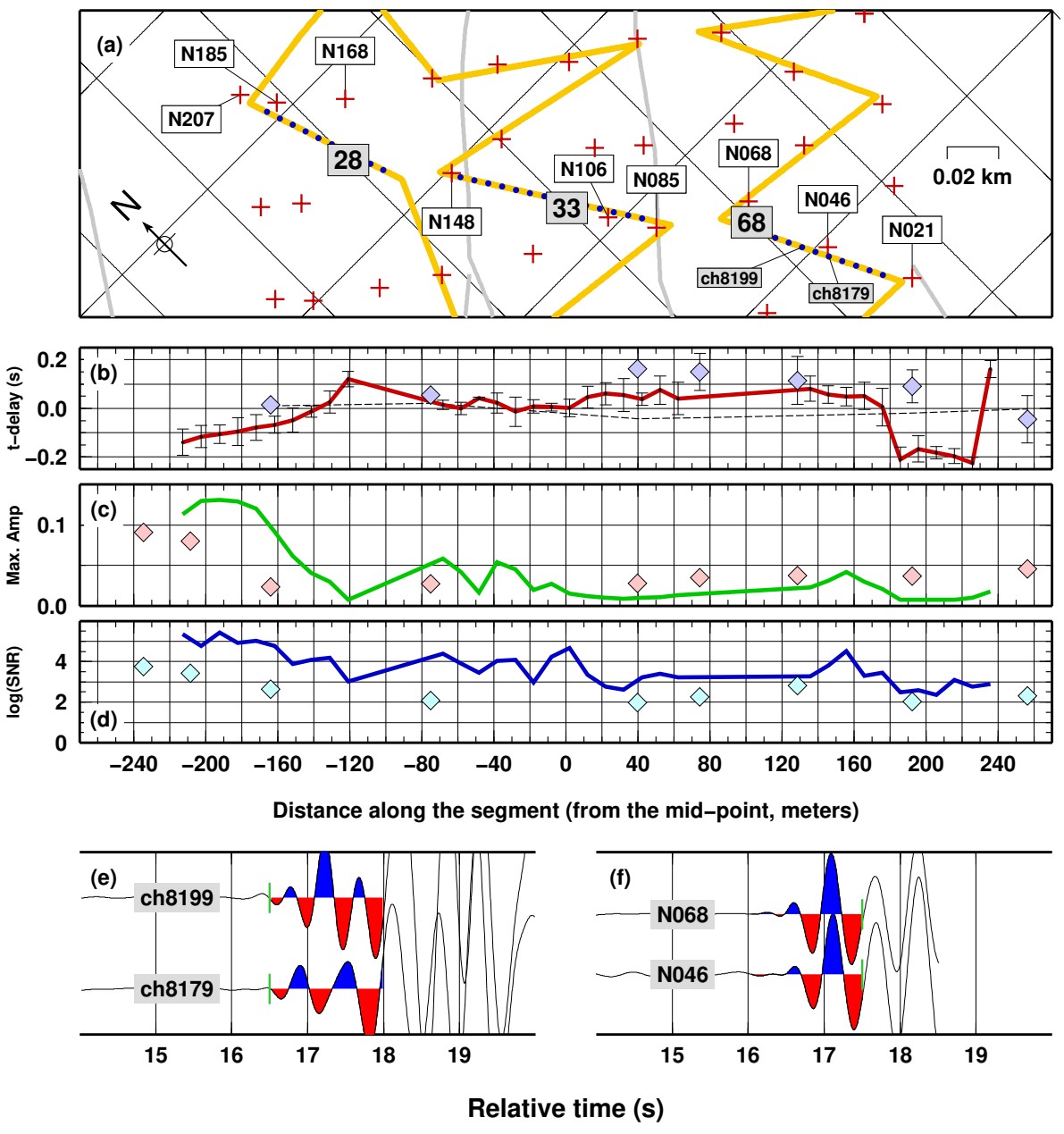

**Figure 8.** Same as Figure 7, but for three consecutive (almost parallel) DAS segments: 68, 33 and 28. (e-f) Examples of waveforms recorded at DAS (e) and nodes (f).

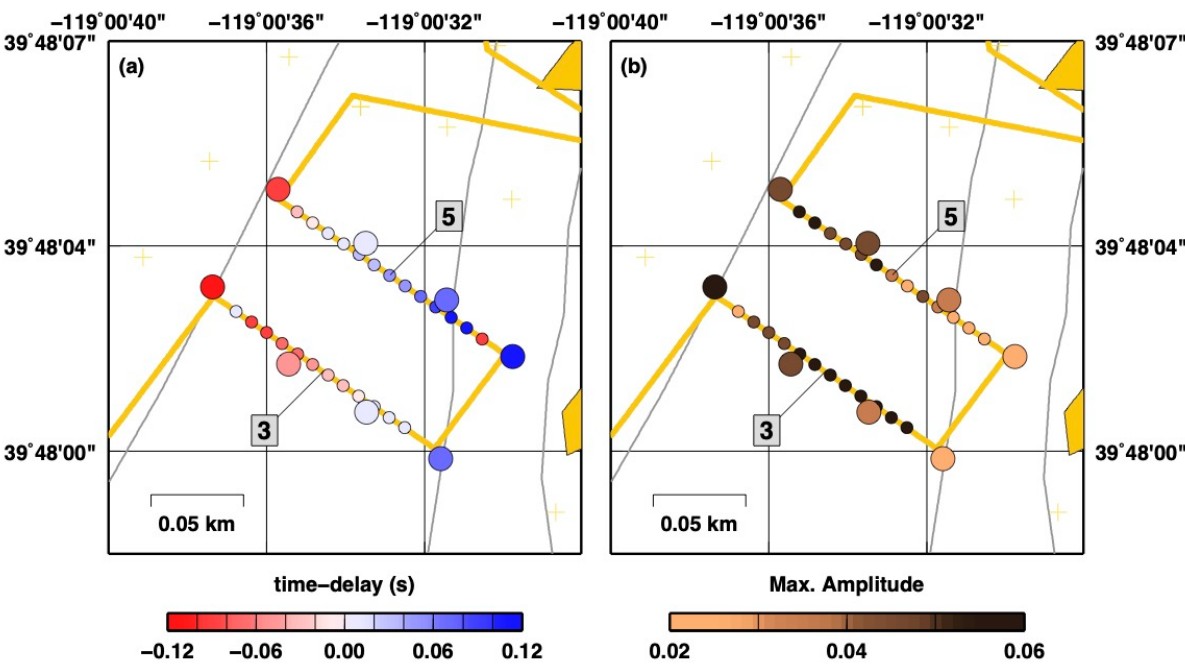

**Figure 9.** Map of: (a) P-wave time-delays and (b) maximum amplitude for two parallel DAS segments: 3 and 5 (small coloured circles). Results for co-located Nodal stations are also presented (large coloured circles).

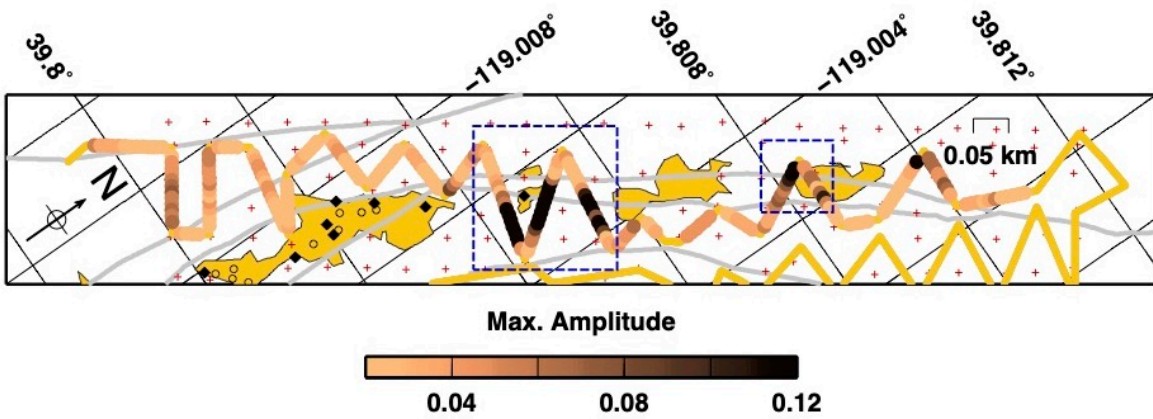

**Figure 10.** Amplitude analysis for all DAS segments along the northwestern side of the Porotomo experiment (segments from from 1 to 30). The dashed box indicate the area of maximum amplitude, which correspond to the fibre cable crossing the warm-ground and the area of the fumaroles/mud-volcanoes.

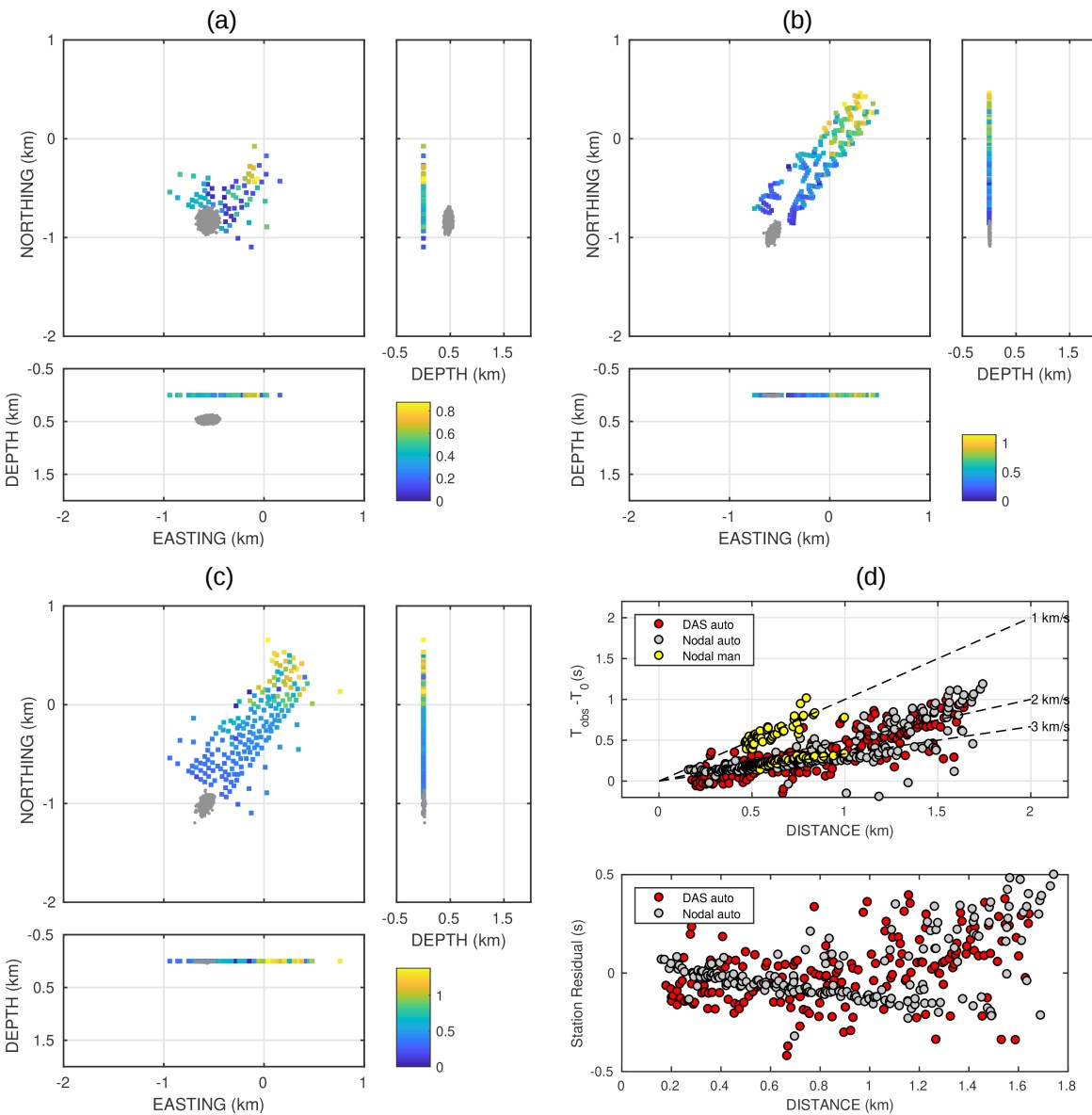

**Figure 11.** Author change: Source location of the earthquake recorded on 2016 March 14 at 10:42:07UTC. (a) Samples of the likelihood function of source location (gray dots) in map view and projected along two vertical sections oriented EW and NS. Results are from travel-time inversion of manually-picked P- and S-wave arrivals at the nodal array. Colors indicate the timing of the picked arrivals, according to the color scale at the bottom right. (b) The same as in (a), but for the automatic picking obtained at the DAS virtual array. The likelihood function for source location is clipped at the surface grid boundary. (c) The same as in (a) and (b), but from inversion of automatically-picked P-wave first arrivals at the nodal array. (d) Arrival times derived at the DAS and nodal arrays as a function of epicentral distance. Dashed lines indicate the move-out corresponding to velocities spanning the [2-4] km/s interval.

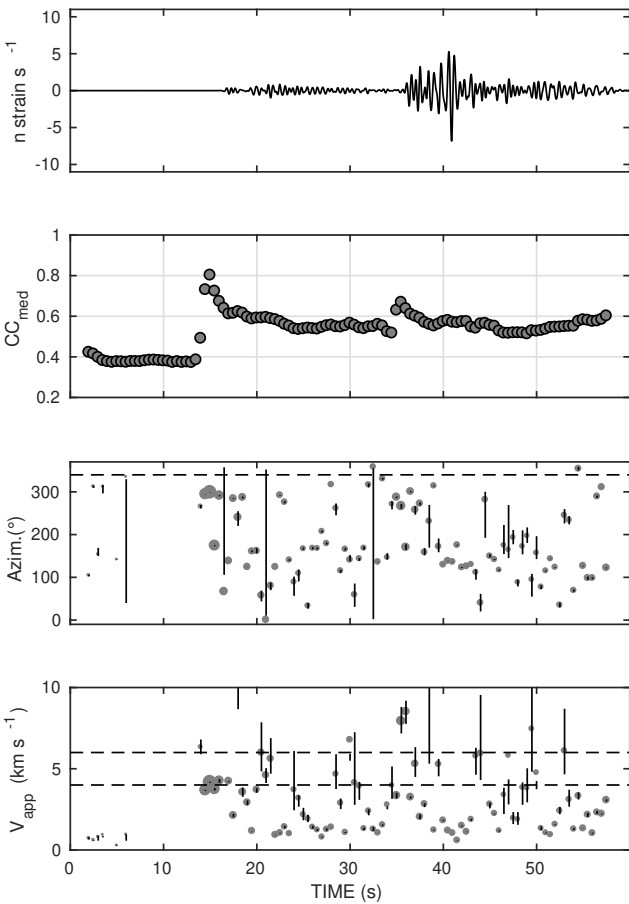

**Figure 12.** Results from fitting a plane wave to DAS differential times. (a) Recording of the Hawthorne earthquake from a sample DAS stacked channel. Data are band-pass filtered over the 0.5-2Hz frequency band. (b) Mean of $C_{ij}^{max}$ from all independent channels of the DAS stacked channels. (c) Propagation azimuth; the size of the points is proportional to the average correlation of the channels used for the inversion. The dashed line indicates the theoretical propagation azimuth ($337°$ clockwise from the N direction). (d) Apparent velocity, derived from the inverse of the modulus of the horizontal slowness vector. The dashed lines mark the velocity range obtained by van den Ende and Ampuero (2021) from analysis of the nodal array data. Vertical bars indicate $2\sigma$ uncertainties in azimuth and velocity.

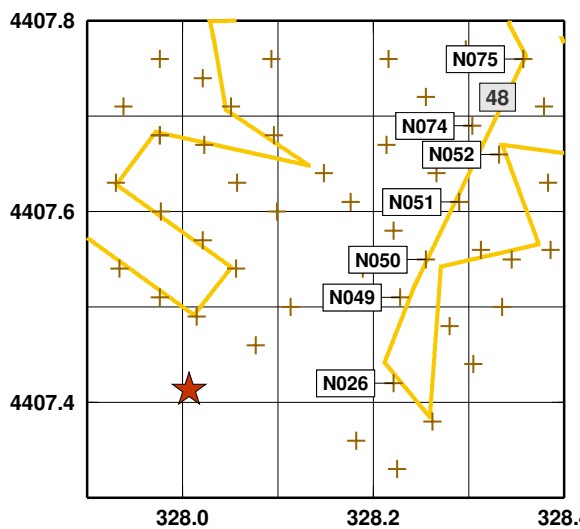

**Figure A1.** Map of the event, nodes and DAS segment used for retrieving P-wave delay-times in the case of a local microseism. Crosses and yellow lines indicate the nodes and the DAS cable, respectively. Investigated nodes and DAS segment are labelled. A red star display the event epicentre, as located using the geophones data and manual pickng of P- and S- phases.

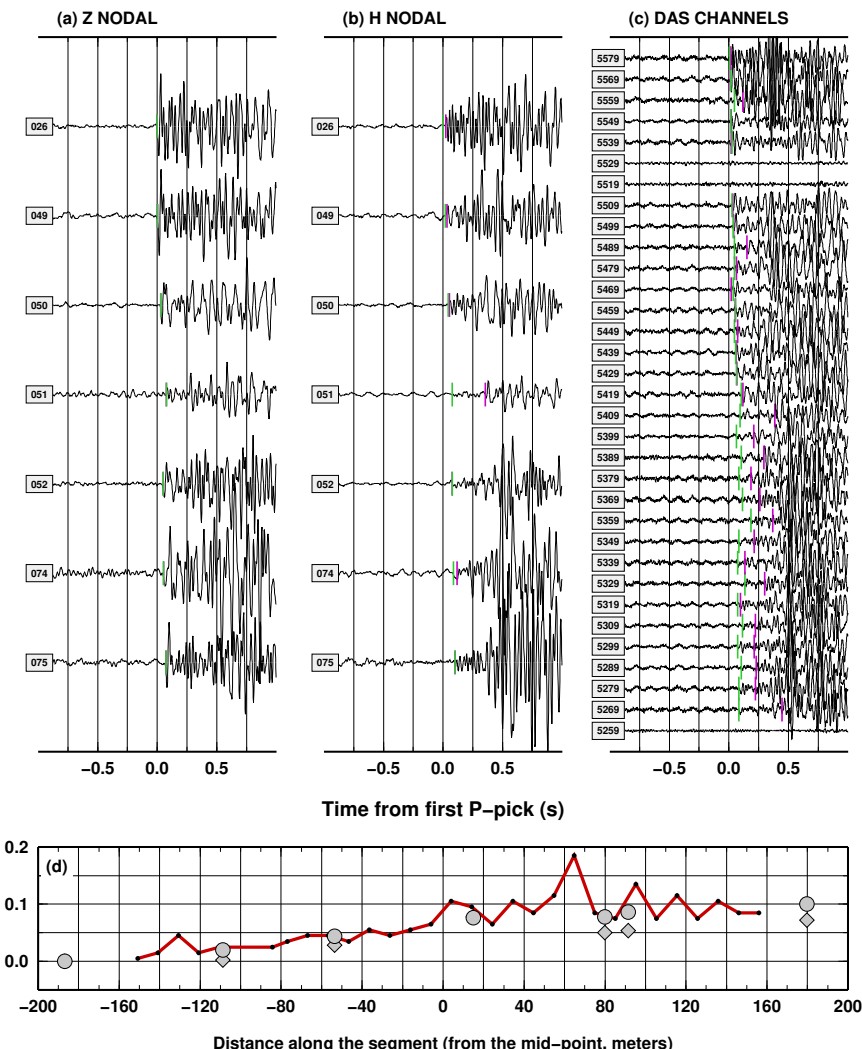

**Figure A2.** (a) Vertical recordings of the geophones along the DAS segment #48. Vertical green and violet bars indicate manual and automatic picking of P-waves. (b) as in (a) but for oriented horizontal component. (c) DAS recordings along segment #48. Symbols as in (a). (d) Relative P-wave delay-times from the first P-pick, for the three set of waves displayed in (a) grey diamonds, (b) grey circles, and (c) red line and black dots.

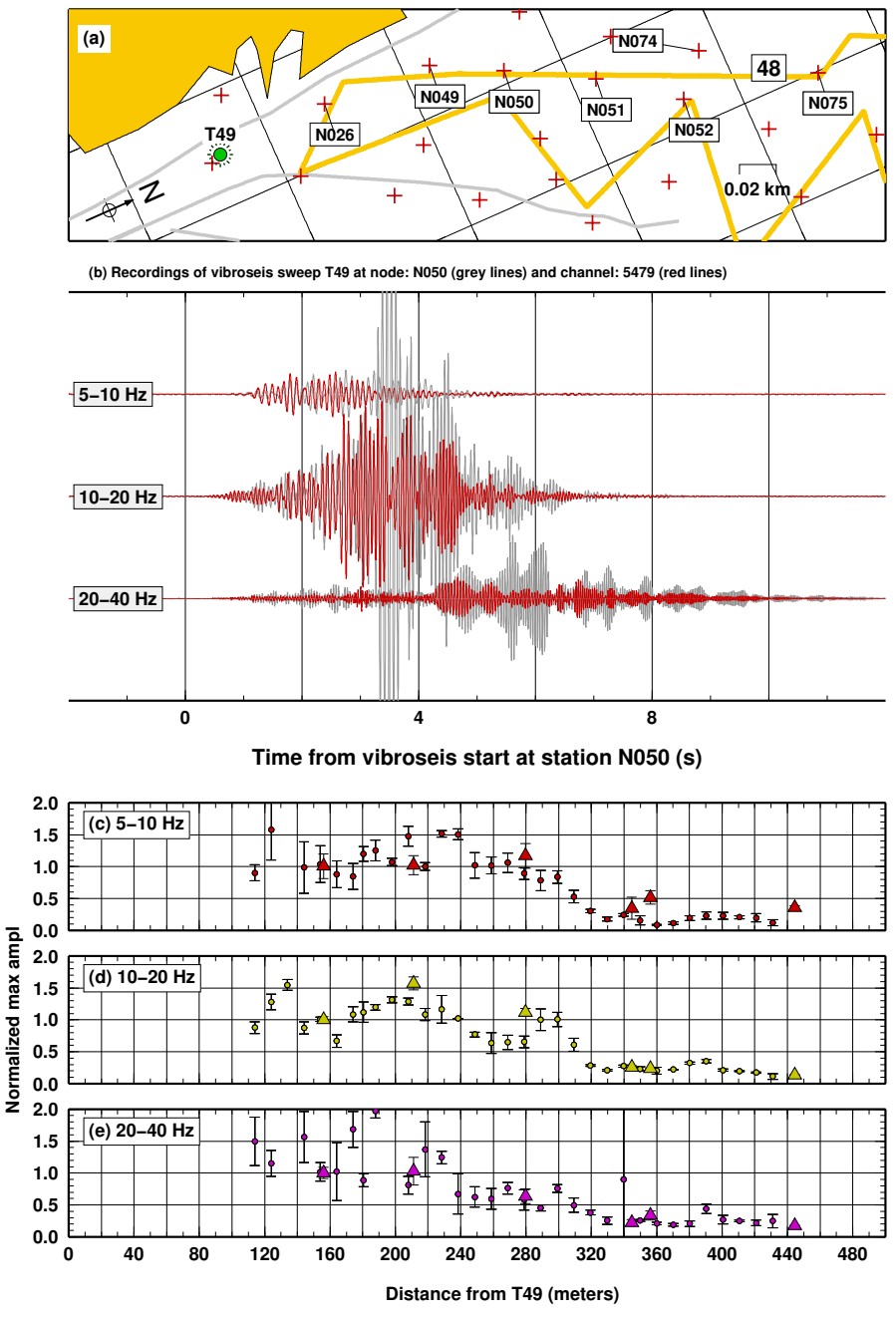

**Figure A3.** Analysis of the amplitude variations along DAS segment #48, from Vibroseis recordings. (a) Map of the Vibroseis source location (grey circle, labelled T49) nodes (labelled grey crosses) and DAS segments #48 (labelled yellow line). (b) Example of filtered recordings of a Vibroseis sweep at co-located node N050 and DAS channel #5479. Variations in maximum amplitude for the nodes (coloured triangles) and the DAS channels (coloured cirlces) in different frequency bands: (c) 5-10 Hz; (d) 10-20 Hz; and (e) 20-40 Hz.