# Peer review of "Distributed acoustic sensing as a tool for subsurface mapping and seismic event monitoring: a proof-of-concept"

_Solid Earth, 2021_

## Referee Comment (RC1)

This study compares a DAS array with a collocated nodal seismometer array in terms of exploration and monitoring performance, described by the authors as (lines 52-54): "For "exploration", we mean the definition of spatial variations of elastic properties in the subsurface at high-resolution. With "monitoring", we have in mind those activities adopted for the real-time detection of "events" (in this case, seismic events)". The authors proceed to detail three procedures for the investigation of the two aforementioned tasks. For the task of exploration, the authors compute relative P-wave arrival times through a cross-correlation method (relative to the mean of the array) for both the DAS and geophone arrays. For the monitoring task, the authors consider one local and one regional earthquake. In the case of the local event, P- and S-arrivals are manually picked from the seismometer data, and DAS P-arrivals are picked automatically. The local earthquake is then located through standard travel time inversion methods. The regional earthquake is analysed by weighted time-domain beamforming of the DAS array.

It is clear that the authors put a lot of effort in these analyses, and they have produced many figures intended to support the claims made in the main text. Unfortunately, not all of these claims are sufficiently well supported by the data, in my opinion. Moreover, the authors do not seem to perform their analyses with the objectives stated in lines 52-54 in mind (no subsurface characterisation, no real-time workflows, no earthquake detection). I would strongly suggest the authors to rethink the focus of this manuscript.

Below I will try to explain my concerns, hoping that the authors recognise my criticism as ways for improvement of the manuscript. Some of my comments may have been a result of misunderstanding, as I found a few sections a bit hard to follow; in that case the authors could simply point this out and make some clarifications.

Kind regards,

Martijn van den Ende

Main comments:

1. **Uncertainties in the relative arrival times**. In Section 3.1, the authors investigate the potential of DAS for exploration, defined as characterising spatial heterogeneities in the phase speed of the medium which manifest themselves in the variations of the relative arrival times. These relative arrival times are estimated through cross-correlation of the waveforms in a 0.5-2 Hz frequency band. The main results of this analysis are presented in Figs. 7 and 8, and the authors discuss trends and deviations therefrom for several DAS segments. When taking Fig. 8 as an example, in which a (non-contiguous) segment of roughly 450m is shown, and assuming an average apparent P-wave speed of 5 km/s (consistent with beamforming analysis), I would expect a maximum difference in relative arrival time of less than 0.09s. However, Fig. 8b shows a variation of up to 0.4s in the DAS array, and about half that in the nodal array. This would suggest to me that the uncertainties in the arrival times are (much) larger than the expected moveout. Potentially the frequency band (up to 2 Hz) is too low for the required precision, or the waveform incoherence introduces additional uncertainties in the arrival time estimation. Regardless, without seeing error bars in Figs. 7 and 8, I am not convinced of any conclusions regarding the comparison

between the DAS and nodal arrays, and the potential for DAS for exploration tasks.

2. **Amplitude variations in relation to site effects**. In the last paragraph of Section 3.1, the authors suggest that the observed variations in the maximum P-wave amplitude are associated with the local geology, attesting to the exploration capacities of DAS. When I look at Fig. 10, I see two regions of elevated amplitudes, but interestingly these do not seem to lie on top of the hydrothermal features indicated in yellow. At other locations where the cable crosses various fault strands or is positioned close to the yellow features, I don't see such a pronounced amplification. And if I had to draw a third box in this figure, I would put it to the far left of the map, where the cable is far away from any geological features. So it would seem to me that the correlation between elevated amplitudes and the local geology is fortuitous at best. At the moment I can't quite think of any analysis that could prove more conclusive in this regard, so perhaps it would be best to take out this paragraph and corresponding figure.

3. **Objectives of Section 3.1**. When I read the authors' definition of "exploration", I initially expected the authors to perform some kind of subsurface imaging similar to what was performed by the PoroTomo team. If my concerns above regarding the uncertainty in the relative arrival times, and the correlation between the waveform amplitude and local geological features are warranted, then Section 3.1 does not offer much in addition to characterise the "spatial variations of elastic properties in the subsurface at high-resolution" (lines 52-53). Instead of relying on 1 passive source (the Hawthorne earthquake), perhaps the authors could use their automated procedure to analyse the numerous vibroseis sweeps that were performed to obtain robust and systematic anomalies in the relative arrival times, or to separate site amplification from the directional sensitivity of DAS, etc. This will likely be a lot of extra work, but it is something that could strengthen Section 3.1 and warrant its existence if the authors agree with my concerns raised in points 1 and 2 above.

4. **Local earthquake location uncertainty**. In Section 3.2.1 the authors locate a local earthquake with manually picked phases recorded on the geophones, yielding a hypocentral depth estimate of 450 +/- 40m. Performing the inversion with EDT on DAS yields a hypocentral depth that is practically at the surface. The authors suggest that "this discrepancy likely owes to the lack of observations attributed to S-waves, and the simplified velocity structure adopted for travel-time predictions" (lines 229-230). These hypotheses can be tested by performing EDT inversion of the nodal array using either only the P-wave picks or both the P- and S-wave picks. If nodal EDT inversion with only the P-arrivals also puts the source at the surface, but not with both the P- and S-arrivals, then the lack of S-wave picks is to blame. Otherwise it is likely that there is an intrinsic problem with EDT inversion of DAS data. This would render DAS inadequate for local earthquake monitoring, and so the claim that "DAS recordings can be used for monitoring and exploration purposes and their performance is at least comparable to seismological records if not superior" (lines 321-322) is unwarranted.

When comparing the hypocentre locations estimated in the present study to those given by Li & Zhan (2018), there appears to be a (very) large discrepancy. Li & Zhan selected 5 earthquakes from the catalogue of Nathwani et al. (2011) as templates for their template matching study. All of the matched events in the catalogue of Li & Zhan should therefore be closely located to the original 5 epicentres, which are all positioned much farther southwest

from the DAS array than shown in Fig. 11 of the present study (placing the inferred epicentres right at the edge of the array). Moreover, the depth of these events was estimated to lie in between 750 and 1250m, and not 450m. So regardless which type of array (geophone or DAS) or inversion method (EDT or absolute) is used, the hypocentre estimates of the authors are very different from the previous estimates. Normally it would be difficult to tell which study is correct, but fortunately the epicentres estimated by the authors are located right next to the instrumented borehole (well 56A-1, 400m deep) at the southern end of the array. From the borehole data the authors should be able to more precisely estimate the depth of the events, and also the distance if the events are really as close as the authors suggested (the propagation front should be strongly curved). So before drawing any conclusions regarding the suitability of DAS for monitoring, the authors should first establish a reliable benchmark to compare their results with. Since the DAS inversion procedure is automated (no manual phase picking), I would also like to see the inversion results of multiple local events, instead of just one (which could be a lucky hit).

5. **Regional earthquake beamforming**. Let me first mention that the manuscript of van den Ende & Ampuero (2020) was recently accepted, and that the version of record is now online. The authors may be interested in reading this revised manuscript, though the revised and newly added sections do not directly pertain to the discussion in the present study. Second, the plane-wave fitting approach (which I think what "PWF" means, but this abbreviation is not defined as far as I can tell) is essentially beamforming in the time domain. In the frequency domain, relative time delays become phase shifts, which form the basis for MUSIC beamforming as used by van den Ende & Ampuero. With MUSIC (and other forms of frequency-domain beamforming) the contribution of each sensor to the overall beampower is weighted by the correlation coefficient similar to Eq. (2) on page 5 of the manuscript. So in essence the PWF method is not very different from frequency-domain beamforming methods. The main difference between this and previous studies would be the stacking procedure that supposedly improves waveform coherence. It would be good to show this directly by comparing the original recordings with the stacked waveforms, and show that the stacking procedure positively affects the coherence/scattering.

Having that said, the authors do indicate that they get better results that previously obtained by vdE&A. Looking at Fig. 12, it is a bit hard to verify this claim, since the scatter in panels c and d is quite large. It would be very helpful to include something like a moving average weighted by the correlation coefficient of the azimuth and the apparent velocity to see the best estimate of these quantities, and their confidence intervals (!). If I were to draw a line by eye, I would put the mean azimuth at around 180 degrees and the apparently velocity at 3 km/s, which are very different from 337 degrees and 4-6 km/s mentioned by the authors.

Lastly, one of the main objectives of this study is to do "real-time detection of events" (lines 52-54), but Section 3.2 does not address either "real-time" or "detection".

6. **Conclusions**. The authors conclude their work with 4 statements, of which I believe are not well supported by the data, or are not a result of this study at all. I would advise the authors to distil a number of take-away messages from their own results.
   a. I've already disputed Statement #1 that DAS is equal or even superior to conventional seismometers for monitoring and exploration.

b. Statement #2 pertains to the volume of DAS data, but this does not make DAS any different from conventional seismic data in terms of the analyses. Other seismological studies (e.g. Roux et al., 2016, GJI) also deal with very dense arrays, and DAS data can always be subsampled if computational resources are limited. Moreover, data volumes and efficiency have not really been the topic of this study.

c. Statement #3 suggests that standard seismological tools cannot be applied to single-component measurements. This is true for polarity analyses, but does not necessarily apply to phase travel time inversions, full-waveform inversion, surface wave tomography, ambient noise interferometry, event detection, beamforming, and possibly numerous other analyses that I forgot to list here. And again, the importance of 1C vs. 3C does not really result from the analyses presented in this study.

d. Statement #4 rehashes previous work on earthquake detection, which was not considered in this study.

Technical comments:

7. In line 120, the authors set-up a velocity model with Vp = 3 km/s. Feigl et al. estimated an average Vp of 2.1 km/s (across all depths), which seems consistent with the beamforming analysis of van den Ende & Ampuero (very crude estimate, though). This may be one reason why the inverted hypocentres are much closer to the array than previously estimated. Also in this line the authors state that they picked 75 vertical and 48 horizontal geophone channels. Shouldn't the picks be the same for all the channels on a given geophone? Why is there a difference in the number of picks? Why did the authors resort to only manual picking for the geophones, and not also do automated picking to assess the quality of the automated picks to compare to the DAS picks?

8. I did not quite follow the stacking procedure (lines 123-126). It is mentioned earlier that only 1 channel is used per gauge, so does "11 adjacent channels are stacked, with a 20-channel step" mean that a stack is created over 11 gauge lengths? Are these stacks of delayed waveforms (aligning the first arrival so that it doesn't get smeared out)? Then in line 126 it is mentioned that the array is downsampled to 274 channels, "similar to the nodal array", but for the nodal array only 75 (+ 48?) stations were picked.

9. Lines 179-180, "Nevertheless, this result supports our workflow where DAS data need to be strictly compared to co-located, re-oriented Nodal data": why is this the case? I don't see how this can be concluded from the preceding paragraph.

10. In Figure 8 I would suggest to add breaks in the red/green/blue lines to indicate where each segment starts and ends, and not showing the gaps in between the segments as measured data.

11. Lines 234-235: but the automated picks should all be for P-waves and their conversions. Was the picking window so large that the direct S-wave could have been picked by mistake?

12. Section 3.2.1: was the origin time co-inverted in the absolute travel time inversion procedure (for the nodal seismometers)? Or was it fixed by the catalogue origin time? EDT doesn't include the origin time, so this information cannot be used to constrain the hypocentre location, which may be another reason why EDT puts the source at the surface. This can be checked by performing EDT on the nodal array.

13. To fully comply with the Open Science philosophy of Solid Earth, could the scripts that are used to process the data be made available in a repository (e.g. Zenodo or Figshare), so that others could reproduce the results?

Minor comments:

- Line 28: "strain" = "particle motion"
- Line 43: the URL to Feigl's department page is not very long-term sustainable, and it will probably turn into a dead link within a few years. Better to refer to the GDR repositories.
- Line 117: typo in the UTC timestamp
- Line 255, tiny detail: van den Ende & Ampuero estimated an apparent P-wave speed of 4-6 km/s (page 919), not 5-6 km/s (although it all falls within the uncertainty).
- Lines 301-302 and 309-310: vdE&A also performed beamforming on subarrays that spanned the entire DAS array (see their Figs. 10-12).
- Lines 305-307: this is actually the other way around. By assuming a single plane-wave, any additional "sources" are discarded to the noise space. By minimising the projection onto the noise space, the effect of scattering is minimised if there is a stronger plane-wave with array-scale coherence. The more sources are assumed, the more the results are biased towards scatterers.
- Lines 311-312: see Lior et al. (10.5194/se-2020-219) for magnitude estimation with DAS.

---

## Referee Comment (RC3)

[referee-annotated manuscript omitted]

---

## Author Comment (AC1)

**Discussion Paper 'Distributed acoustic sensing as a tool for exploration and monitoring: a proof-of-concept', by Nicola Piana Agostinetti and co-workers.**

**Response to Reviewer #1 (Martijn van den Ende)**

**Reviewer #1**

This study compares a DAS array with a collocated nodal seismometer array in terms of exploration and monitoring performance, described by the authors as (lines 52-54): "For "exploration", we mean the definition of spatial variations of elastic properties in the subsurface at high-resolution. With "monitoring", we have in mind those activities adopted for the real-time detection of "events" (in this case, seismic events)". The authors proceed to detail three procedures for the investigation of the two aforementioned tasks. For the task of exploration, the authors compute relative P-wave arrival times through a cross-correlation method (relative to the mean of the array) for both the DAS and geophone arrays. For the monitoring task, the authors consider one local and one regional earthquake. In the case of the local event, P- and S-arrivals are manually picked from the seismometer data, and DAS P-arrivals are picked automatically. The local earthquake is then located through standard travel time inversion methods. The regional earthquake is analysed by weighted time-domain beamforming of the DAS array.

It is clear that the authors put a lot of effort in these analyses, and they have produced many figures intended to support the claims made in the main text. Unfortunately, not all of these claims are sufficiently well supported by the data, in my opinion. Moreover, the authors do not seem to perform their analyses with the objectives stated in lines 52-54 in mind (no subsurface characterisation, no real-time workflows, no earthquake detection). I would strongly suggest the authors to rethink the focus of this manuscript.

We thank Reviewer #1 for pointing out this flaw in the general overview of the manuscript. We believe that the goals of our study are not clearly explained in the manuscript and they could have been mis-understood (see also Reviewer #2's comments). Moreover, we feel that there is a mis-conception in how we reached these goals. For example, "subsurface characterization" is not our goal (e.g. building a velocity model or recovering subsurface structures), because such task could be done in many different ways and different data could need different algorithms/approaches. Here we are reporting how the relevant data (in case of "subsurface reconstruction" are the P-wave travel times) extracted from DAS recordings compare to the same quantities extracted from "node" recordings. We will make this point crystal clear in the introduction.

Below I will try to explain my concerns, hoping that the authors recognise my criticism as ways for improvement of the manuscript. Some of my comments may have been a result of misunderstanding, as I found a few sections a bit hard to follow; in that case the authors could simply point this out and make some clarifications.

Kind regards,

Martijn van den Ende

Main comments:

**1. Uncertainties in the relative arrival times**. In Section 3.1, the authors investigate the potential of DAS for exploration, defined as characterising spatial heterogeneities in the

phase speed of the medium which manifest themselves in the variations of the relative arrival times. These relative arrival times are estimated through cross-correlation of the waveforms in a 0.5-2 Hz frequency band. The main results of this analysis are presented in Figs. 7 and 8, and the authors discuss trends and deviations therefrom for several DAS segments. When taking Fig. 8 as an example, in which a (non-contiguous) segment of roughly 450m is shown, and assuming an average apparent P-wave speed of 5 km/s (consistent with beamforming analysis), I would expect a maximum difference in relative arrival time of less than 0.09s. However, Fig. 8b shows a variation of up to 0.4s in the DAS array, and about half that in the nodal array. This would suggest to me that the uncertainties in the arrival times are (much) larger than the expected moveout. Potentially the frequency band (up to 2 Hz) is too low for the required precision, or the waveform incoherence introduces additional uncertainties in the arrival time estimation. Regardless, without seeing error bars in Figs. 7 and 8, I am not convinced of any conclusions regarding the comparison between the DAS and nodal arrays, and the potential for DAS for exploration tasks.

We will add uncertainties in the P-travel-times obtained with a Bayesian approach (see Piana Agostinetti and Martini, Sedimentary basins investigation using teleseismic P-wave time delays, Geophysical prospecting, 2019) to make our discussion more focused on the point raised by the Reviewer.

**2. Amplitude variations in relation to site effects**. In the last paragraph of Section 3.1, the authors suggest that the observed variations in the maximum P-wave amplitude are associated with the local geology, attesting to the exploration capacities of DAS. When I look at Fig. 10, I see two regions of elevated amplitudes, but interestingly these do not seem to lie on top of the hydrothermal features indicated in yellow. At other locations where the cable crosses various fault strands or is positioned close to the yellow features, I don't see such a pronounced amplification. And if I had to draw a third box in this figure, I would put it to the far left of the map, where the cable is far away from any geological features. So it would seem to me that the correlation between elevated amplitudes and the local geology is fortuitous at best. At the moment I can't quite think of any analysis that could prove more conclusive in this regard, so perhaps it would be best to take out this paragraph and corresponding figure.

Looking at Figure 10, it is clear that the quantity "MaxAmp" reaches consistently (i.e. the same MaxAmp value for more consecutive DAS channels) large values (>0.10) only in the two boxes we overplotted, and there is no other area with such high values. Clearly, such threshold (0.10) is subjective and could be discussed.  We agree with the Reviewer that there are other areas where the DAS cable is close to the "warm ground" (yellow areas), without traversing it and without showing such large MaxAmp values. We will add a comment about this in the manuscript.

**3. Objectives of Section 3.1**. When I read the authors' definition of "exploration", I initially expected the authors to perform some kind of subsurface imaging similar to what was performed by the PoroTomo team. If my concerns above regarding the uncertainty in the relative arrival times, and the correlation between the waveform amplitude and local geological features are warranted, then Section 3.1 does not offer much in addition to characterise the "spatial variations of elastic properties in the subsurface at high-resolution" (lines 52-53). Instead of relying on 1 passive source (the Hawthorne earthquake), perhaps the authors could use their automated procedure to analyse the numerous vibroseis sweeps that were performed to obtain robust and systematic anomalies in the relative arrival times,

or to separate site amplification from the directional sensitivity of DAS, etc. This will likely be a lot of extra work, but it is something that could strengthen Section 3.1 and warrant its existence if the authors agree with my concerns raised in points 1 and 2 above.

As we clarified at the beginning of this letter, we think there is a bit of mis-understanding about the "exploration" concept and what the Reviewer was expecting. For example, we are not interested here "to perform some kind of subsurface imaging" because, as we know from some decades of studies in such a field, imaging is 80% the algorithm you use to do it. Here we are interested to see how the relevant data, extracted from DAS recordings, compare to the same data, extracted from "node" recordings. It could be useful to study which imaging algorithm would give the best result with DAS data (given the geometry of the DAS cable compared to the geometry of standard geophone networks, the directionality of the recordings and so on), but this is far beyond the scope of our study.

**4. Local earthquake location uncertainty.** In Section 3.2.1 the authors locate a local earthquake with manually picked phases recorded on the geophones, yielding a hypocentral depth estimate of 450 +/- 40m. Performing the inversion with EDT on DAS yields a hypocentral depth that is practically at the surface. The authors suggest that "this discrepancy likely owes to the lack of observations attributed to S-waves, and the simplified velocity structure adopted for travel-time predictions" (lines 229-230). These hypotheses can be tested by performing EDT inversion of the nodal array using either only the P-wave picks or both the P- and S-wave picks. If nodal EDT inversion with only the P-arrivals also puts the source at the surface, but not with both the P- and S-arrivals, then the lack of S-wave picks is to blame. Otherwise it is likely that there is an intrinsic problem with EDT inversion of DAS data. This would render DAS inadequate for local earthquake monitoring, and so the claim that "DAS recordings can be used for monitoring and exploration purposes and their performance is at least comparable to seismological records if not superior" (lines 321-322) is unwarranted.

Table 1

| Test # | Deployment | Picking | Method | Wave type | Vp | x | y | z |
|--------|------------|---------|--------|-----------|------|-----------|-----------|------------|
| 1 | NODAL | man | GAU | P,S | 2000 | -0.485937 | -0.692187 | 0.308301 |
| 2 | DASH | auto | EDT | - | 2000 | -0.448437 | -0.717187 | 0.204199 |
| 3 | DASH | auto | GAU | - | 2000 | -0.321875 | -1.99688 | 0.00800781 |
| 4 | DASH | auto | EDT | - | 3000 | -0.476562 | -0.767187 | 0.00400391 |
| 5 | DASH | auto | GAU | - | 3000 | -1.04688 | -1.99688 | 0.00800781 |
| 6 | NODAL | auto | EDT | P | 3000 | -0.489062 | -0.751562 | 0.23623 |

| 7 | **NODAL** | **man** | **GAU** | **P,S** | **3000** | **-0.576562** | **-0.851562** | **0.460449** |
|---|---|---|---|---|---|---|---|---|
| 8 | NODAL | man | EDT | P,S | 3000 | -0.532812 | -0.798437 | 0.436426 |
| 9 | NODAL | man | EDT | P | 3000 | -0.495312 | -0.732812 | 0.252246 |
| 10 | NODAL | auto | GAU | P | 3000 | -0.61875 | -0.93125 | 0.720703 |

We performed the test requested by the reviewer (see Table 1). In substance, we found that: (1) The EDT approach applied to manual P- and S-wave pickings from the nodal array (Test #8) does not yield any substantial modification in hypocentral coordinates, including depth; (2) By repeating the EDT inversion using only P-wave readings, obtained either automatically (Test #6) or manually (Test #9), the hypocentral solution gets a bit closer to the deployment, at a shallower depth (236-252 m vs 460 of the reference solution). Thus, it looks like S-wave readings are more relevant than the location algorithm in conditioning the depth estimates.
However, a full understanding of the dependence of source location on arrival times data, velocity structure and location algorithm would require a thorough assessment of the topology of the misfit function, a task which is beyond the purpose of the present paper. To conclude, we agree with the reviewer that these results do not imply a good performance of DAS recordings for the location of local microearthquakes; the conclusions will be modified consequently

When comparing the hypocentre locations estimated in the present study to those given by Li & Zhan (2018), there appears to be a (very) large discrepancy. Li & Zhan selected 5 earthquakes from the catalogue of Nathwani et al. (2011) as templates for their template matching study. All of the matched events in the catalogue of Li & Zhan should therefore be closely located to the original 5 epicentres, which are all positioned much farther southwest from the DAS array than shown in Fig. 11 of the present study (placing the inferred epicentres right at the edge of the array).
This statement is questionable. In building their catalog of matched events, Li and Zhan (2018) also included events whose correlation with the respective template waveform was very low. For the specific earthquake analysed in the present work, the peak correlation value was as low as 0.11 (see Table S1 in the Supplemental Material of Li and Zhan, 2018). As a consequence, the closeness between template and matched hypocenters is not necessarily implied.

Moreover, the depth of these events was estimated to lie in between 750 and 1250m, and not 450m. So regardless which type of array (geophone or DAS) or inversion method (EDT or absolute) is used, the hypocentre estimates of the authors are very different from the previous estimates.
See the reply posted above; the low correlation coefficient of the analysed earthquake does not demonstrate its closeness to the template events. Moreover: we're not particularly concerned about the absolute location of these events; rather, we're interested in the

comparison between the location obtained by inverting manually-picked P- and S-wave arrivals at the geophone array (here taken as a reference), and the DAS solution.

Normally it would be difficult to tell which study is correct, but fortunately the epicentres estimated by the authors are located right next to the instrumented borehole (well 56A-1, 400m deep) at the southern end of the array. From the borehole data the authors should be able to more precisely estimate the depth of the events, and also the distance if the events are really as close as the authors suggested (the propagation front should be strongly curved).

We may be wrong, but it looks like the vertical DAS cable located within well 56A-1 was not operative during the day of the local earthquake (see DASV metadata at https://gdr.openei.org/files/829/DAS_Borehole_cable_files.txt).

So before drawing any conclusions regarding the suitability of DAS for monitoring, the authors should first establish a reliable benchmark to compare their results with. Since the DAS inversion procedure is automated (no manual phase picking), I would also like to see the inversion results of multiple local events, instead of just one (which could be a lucky hit).

This is a pertinent observation; in the revised MS we'll take care of analysing additional earthquakes to better corroborate our conclusions, and to verify the performance of the entire workflow on a larger data set.

**5. Regional earthquake beamforming.** Let me first mention that the manuscript of van den Ende & Ampuero (2020) was recently accepted, and that the version of record is now online. The authors may be interested in reading this revised manuscript, though the revised and newly added sections do not directly pertain to the discussion in the present study. Second, the plane-wave fitting approach (which I think what "PWF" means, but this abbreviation is not defined as far as I can tell) is essentially beamforming in the time domain. In the frequency domain, relative time delays become phase shifts, which form the basis for MUSIC beamforming as used by van den Ende & Ampuero. With MUSIC (and other forms of frequency-domain beamforming) the contribution of each sensor to the overall beampower is weighted by the correlation coefficient similar to Eq. (2) on page 5 of the manuscript. So in essence the PWF method is not very different from frequency-domain beamforming methods. The main difference between this and previous studies would be the stacking procedure that supposedly improves waveform coherence. It would be good to show this directly by comparing the original recordings with the stacked waveforms, and show that the stacking procedure positively affects the coherence/scattering. Having that said, the authors do indicate that they get better results that previously obtained by vdE&A. Looking at Fig. 12, it is a bit hard to verify this claim, since the scatter in panels c and d is quite large. It would be very helpful to include something like a moving average weighted by the correlation coefficient of the azimuth and the apparent velocity to see the best estimate of these quantities, and their confidence intervals (!). If I were to draw a line by eye, I would put the mean azimuth at around 180 degrees and the apparently velocity at 3 km/s, which are very different from 337 degrees and 4-6 km/s mentioned by the authors.

Before answering these comments, we wish to clarify a basic aspect. With our re-processing of the Hawthorne earthquake using DAS as a sensor array, it was absolutely not in our intention to question van den Ende & Ampuero (2020) results. Rather, this latter work

provided the impetus and the inspiration for our efforts. Having clarified this issue, let's proceed with a point-by-point reply to the above issues.

(1) Yes, the reviewer is right: PWF stands for Plane Wave Fitting. We apologize for not having made that acronym explicit; It will be fixed in the revised MS.

(2) PWF is **substantially different** from beam-forming (either in the time- or frequency-domain) methods. In PWF, inter-channel time delays are first estimated via cross-correlation, and then used to solve a linear problem in which the kernel matrix is given by the differential array coordinates, and the model parameters are the two (or three, in case of a 3D array) component of the slowness vector. Conversely, beamforming methods are based upon a time (or phase) shifting according to a tentative slowness, and on the successive calculation of the beam power for those time- (phase-) shifted time series (spectra). Thus, the differences between the two approaches are substantial. In the first case, we have a linear problem which is directly solved using least squares, while in the second case the problem is non linear and it is approached through (a) solving the forward problem of predicting time delays (phase shifts) given a slowness vector, and (b) using those prediction to calculate a slant-stack, or beam power, of the time- (phase-) shifted seismograms (spectra). The procedure is iterated using a direct search over the slowness plane, and the solution is given by that slowness for which the power of the beam is maximized.

Having clarified this, it is difficult to tell which technique works better under which conditions. In principle, methods exploiting the properties of the spatial covariance matrix of the signal such as MUSIC are expected to provide superior performances in the case of multiple, interfering signals.  On the other hand, these latter approaches do not allow to comprehensively account for the coherence between individual channel pairs (unless the covariance matrix is normalized), or to exclude from the slowness estimate channel pairs which are poorly correlated.

Lastly, one of the main objectives of this study is to do "real-time detection of events" (lines 52-54), but Section 3.2 does not address either "real-time" or "detection".

Actually, at pg. 8, lines 250-253, we observe that '*As a matter of fact, the onset of the earthquake signal is marked by an abrupt increase of the average multi channel correlation (Figure 12b) which, in correspondence of the P-wave arrival, peaks to a value which is about 2 times larger than those associated with the preceding background noise*'. Then, at lines 260-261, we conclude that: '....*a simple thresholding on the overall correlation of DAS channels may serve as an efficient operator for detecting the arrival of an earthquake signal*'.

However: we recognize that we should have stressed this point better, so that in the revised MS some additional statements will be reported.

6. Conclusions. The authors conclude their work with 4 statements, of which I believe are not well supported by the data, or are not a result of this study at all.

That's a rather hard statement! Why such an acrimonious treatment ?

I would advise the authors to distil a number of take-away messages from their own results.

We'll do our best, promise.

a. I've already disputed Statement #1 that DAS is equal or even superior to conventional seismometers for monitoring and exploration.

We partially agree on this. By the time of comparing DAS to conventional seismometer deployments, one has also to account for the ease of deployment and maintenance of an equivalent number of channels. Under this perspective, we believe that DAS permits faster and easier installation and maintenance procedures. This point will be clarified in the revised MS.

b. Statement #2 pertains to the volume of DAS data, but this does not make DAS any different from conventional seismic data in terms of the analyses. Other seismological studies (e.g. Roux et al., 2016, GJI) also deal with very dense arrays, and DAS data can always be subsampled if computational resources are limited. Moreover, data volumes and efficiency have not really been the topic of this study.

The referenced statement is a bit more complex. First, thousands of channels sampled at 1KHz do not really represent a standard in passive seismology practice. Other than computational time and storage issues, such large amount of data also need spatial/temporal subsampling and, as described in our and several other papers, a selection aimed at removing data which may be considered 'bad' a priori (e.g., channels in proximity of the fiber's corners). But this introduces a further issue, that concern the a-priori discrimination between 'bad' and 'good' data…
So: we recognise that our statement #2 was definitely too synthetic, and we will make an effort to clarify the arguments there reported.

c. Statement #3 suggests that standard seismological tools cannot be applied to single-component measurements. This is true for polarity analyses, but does not necessarily apply to phase travel time inversions, full-waveform inversion, surface wave tomography, ambient noise interferometry, event detection, beamforming, and possibly numerous other analyses that I forgot to list here. And again, the importance of 1C vs. 3C does not really result from the analyses presented in this study.

We thank the reviewer for the kind reminder on the basic methods of observational seismology. Unfortunately, we still guess that a DAS deployment may exhibit limitations in the measurement of waves whose polarization vector has poor or null projection along the elongation of the fiber. So: no matter the number of possible applications using DAS data, their effectiveness will always depend on the relationships between the wave direction of propagation and polarization, and the elongation of the sensing fiber. As a matter of fact, inspection of Fig. 11 in the original MS evidences that the automatic phase picker identify as a first break either the P- or S-wave arrivals, depending on the orientation of the fiber with respect to the back azimuth, which in turn condition which of the two different wave types is recorded best.

d. Statement #4 rehashes previous work on earthquake detection, which was not considered in this study.

This criticism is not very clear; we'll try to better explain statement #4. By the time of evaluating DAS performances for earthquake monitoring, an issue is represented by the fact that it may be difficult for a single fiber to achieve a complete azimuthal coverage of seismogenic regions extended over areas of hundreds of squared kilometers or more. Since the DAS is inherently a multichannel system, a given seismogenic area could be monitored by multiple DASs, which would thus constitute an *array of arrays.* Under this perspective, a better location procedure would consist in the back-propagation of the wave-vectors evaluated at individual DAS systems.

Technical comments:
7. In line 120, the authors set-up a velocity model with Vp = 3 km/s. Feigl et al. estimated an average Vp of 2.1 km/s (across all depths), which seems consistent with the beamforming analysis of van den Ende & Ampuero (very crude estimate, though).

Since we were principally interested in the relative locations between the nodal and DAS deployments, for the sake of simplicity we adopted a homogeneous velocity structure whose velocity was chosen by trial-and-error, i.e. by repeating the locations using a velocity of the half-space in between 2 and 5 km/s, and using the RMS of station residuals as a goodness-of-fit indicator. As for the reference to Feigl et al.: we deem the citation pertains to the work 'PoroTomo Final Technical Report:Poroelastic Tomography by Adjoint Inverse Modeling of Data from Seismology, Geodesy, and Hydrology', with digital identifier https://doi.org/10.2172/1499141. If that's the case, in the tomographic slices shown in Fig. 5.5, we observe P-wave velocity increasing rapidly from ~1000 m/s at the surface to 3000 m/s or more at depths of 200-250 m. So, for a source located at depths of several hundred meters (or even 1700m, as in the case of Li & Zhan template events; see response to point above), the 3000 m/s choice does not appear to be so exotic.

This may be one reason why the inverted hypocentres are much closer to the array than previously estimated. Also in this line the authors state that they picked 75 vertical and 48 horizontal geophone channels. Shouldn't the picks be the same for all the channels on a given geophone? Why is there a difference in the number of picks?

In seismological observatory practice, it occurs very often that a given station has a very clear P-wave onset, but much noisier/unclear S-wave arrivals. When, such in our case, a huge number of recordings is available, there's no point in measuring phase arrival times with large uncertainties; as a consequence, the number of S-wave readings may be different from that of P-waves.

Why did the authors resort to only manual picking for the geophones, and not also do automated picking to assess the quality of the automated picks to compare to the DAS picks?

As clearly stated in the manuscript, the point of using the nodal data was to obtain a reference location for assessing the performance of automatic DAS processing. That's why

data from the nodal array were picked manually, while leaving the automatic approach to the DAS recordings.
In any case, we made automatic picking on nodal data as well, so to compare the locations from the two deployments under comparable conditions (see Table 1, Tests #4 and #6). The two locations are basically the same for which concerns the two horizontal coordinates, but differs significantly for the depth one (~0 for the DAS, and ~230m for the nodal array).

8. I did not quite follow the stacking procedure (lines 123-126). It is mentioned earlier that only 1 channel is used per gauge, so does "11 adjacent channels are stacked, with a 20-channel step" mean that a stack is created over 11 gauge lengths? Are these stacks of delayed waveforms (aligning the first arrival so that it doesn't get smeared out)? Then in line 126 it is mentioned that the array is downsampled to 274 channels, "similar to the nodal array", but for the nodal array only 75 (+ 48?) stations were picked.

The stacking procedure adopted in the 'exploration' part is different from that used in the 'monitoring' one. For this latter case, we performed exactly what is written in the text, by stacking (for instance) channels 1-11,  21-31, etc…, then assigning to the stacked waveform the coordinates of channels 6 and 26, respectively.

9. Lines 179-180, "Nevertheless, this result supports our workflow where DAS data need to be strictly compared to co-located, re-oriented Nodal data": why is this the case? I don't see how this can be concluded from the preceding paragraph.

10. In Figure 8 I would suggest to add breaks in the red/green/blue lines to indicate where each segment starts and ends, and not showing the gaps in between the segments as measured data.

Agreed

11. Lines 234-235: but the automated picks should all be for P-waves and their conversions. Was the picking window so large that the direct S-wave could have been picked by mistake?

This is a pertinent comment. Our automatic picking was performed using a window containing the entire event waveform (1 minute, in this case). The picks were later associated using the DBSCAN clustering algorithm, following the procedure adopted by the ScanLoc module (https://docs.gempa.de/scanloc/current/#confval-clusterSearch.maxSearchDist).
So, yes: the reviewer is right.  From inspection of Figure 11, panel (d) it appears that many pickings are in between the expected P- and S-wave travel times, and just a few picks are consistent with S-wave arrivals. The statement '*Many onsets, therefore, are likely representative of S-wave arrivals, following either direct or scattered paths* ' will be changed to '*Many onsets, therefore, are likely representative of late arrivals from the P-wave coda*'.

12. Section 3.2.1: was the origin time co-inverted in the absolute travel time inversion procedure (for the nodal seismometers)? Or was it fixed by the catalogue origin time? EDT

doesn't include the origin time, so this information cannot be used to constrain the hypocentre location, which may be another reason why EDT puts the source at the surface. This can be checked by performing EDT on the nodal array.

In both the 'classic' and EDT location methods, the origin time is calculated *a posteriori* using the travel-times to the maximum-likelihood hypocenter. The tests performed for answering to point #4 above suggest that the lack of S-wave arrivals is more relevant than EDT in putting the source at the surface.

13. To fully comply with the Open Science philosophy of Solid Earth, could the scripts that are used to process the data be made available in a repository (e.g. Zenodo or Figshare), so that others could reproduce the results?
For sure. All the relevant scripts will be made available by the time of submitting the revised version of the MS.

Minor comments:

Line 28: "strain" = "particle motion"

Line 43: the URL to Feigl's department page is not very long-term sustainable, and it will probably turn into a dead link within a few years. Better to refer to the GDR repositories.

Line 117: typo in the UTC timestamp

Line 255, tiny detail: van den Ende & Ampuero estimated an apparent P-wave speed of 4-6 km/s (page 919), not 5-6 km/s (although it all falls within the uncertainty).

Lines 301-302 and 309-310: vdE&A also performed beamforming on subarrays that spanned the entire DAS array (see their Figs. 10-12).

Lines 305-307: this is actually the other way around. By assuming a single plane-wave, any additional "sources" are discarded to the noise space. By minimising the projection onto the noise space, the effect of scattering is minimised if there is a stronger plane-wave with array-scale coherence. The more sources are assumed, the more the results are biased towards scatterers.

Lines 311-312: see Lior et al. (10.5194/se-2020-219) for magnitude estimation with DAS.

---

## Author Comment (AC2)

**Discussion Paper 'Distributed acoustic sensing as a tool for exploration and monitoring: a proof-of-concept', by Nicola Piana Agostinetti and co-workers.**

**Response to Reviewer #2**

**Reviewer #2**

This paper is a relatively brief focused discussion on methods for how a DAS array can be used to 1) map subsurface heterogeneity ("exploration") and 2) detect and locate seismic events ("monitoring"). The methods are validated and compared with geophone ("node") data co-located with the array in the publicly available PoroTomo data set. The paper achieves its goals and is a useful contribution to understanding how to best to use DAS data. The paper should be particularly valuable to anyone planning a surface DAS array for one or both of the targeted applications. While I recommend publication, I do have a list of editorial comments, suggestions, and questions.

The most significant one is for readability and it is to consider combining the methods and results for the two categories of results ("exploration" and "monitoring"). I do also prefer that "exploration" be called "subsurface site effects" or "subsurface heterogeneity mapping" and the "monitoring" be called "seismic event and location detection". These titles are clumsy but they don't overpromise. After all, the basic result in the "exploration" category is relative site effect time delays of +/- 0.12 seconds.

As discussed in the response letter to Reviewer #1's comments, we think that we have to clarify the manuscript goals and how we reach them. We think that "exploration" and "monitoring" should be more appropriate terms, but we also know that such terms have been used for deeper targets ("exploration") and wider workflows ("monitoring") than what we did present here. We will try to clarify goals and terminology as much as possible in the revised manuscript.

Therefore, I might also suggest a more specific title, e.g., Methods for subsurface mapping and event detection using Distributed Acoustic Sensing: Examples from Brady Hot Springs PoroTomo Experiment

We agree with the suggested title and we will change it to make the manuscript more appealing

Specific Comments, Suggestions, Questions.
l. 1. "PoroTomo" is short for "Poroelastic Tomography" so there is no reason to have the "TOMO" all caps.

l. 36 - Marra et al. is not DAS and is not relevant to this paper.

l. 103-104 - No node instrument response corrections were made. Is that relevant?

l. 115-117 – Given over 100 events in the Li and Zhan catalogue, maybe give a liitle more reason for choosing the 14 March 10:41 UTC event. Is it the closest to the array?

Yes, that event was selected since, according to a preliminary inspection of arrival times, it appeared to be located in close proximity of the deployment, and it exhibited good visibility in

both waveforms and cross-correlations (see Supplementary Materials in Li & Zhan, 2018). Anyhow, in the revised MS we will present results for more events in the revised manuscript.

l. 118-150 – These paragraphs from "Methods" would most seamlessly be followed by the "Location" results (see l. 219).l. 124-125- Is the meaning of "20-channel step" that channels 1-11, 31-41, 61-71, etc. are stacked if Channels 1-100 are a segment?

l. 179-180 Be explicit that "strictly compared" means that nodal horizontal components need to be vector summed in DAS direction.

L.207-209. Can you suggest a reason why the criterion for removing channels needs to be increased to 2 gage lengths from a corner?
There's no particular reason behind this choice. It represented a conservative approach toward channel selection.

l. 212-216 – Can you relate this result to local site effects presented in Parker et al. or Zeng et al.?

l. 219 – Note how out-of-context this statement is because its antecedent goes back to l.

118-135. It is best to discuss this result right after l. 135. Even though the paper is short, I don't think the reader can be expected to keep this in mind. Consider keeping the "data and methods" and their corresponding results together.

l. 225 – Similarly to above, EDT was referred to back in "methods" section. Also, "Conversely" is a term of logic. The meaning here is "In comparison".

l. 233-234 - "projection" implies vector, not tensor, projection. $\cos^2$ is stronger than $\cos$.

l. 250 - "PWF" Plane-wave fitting is never defined.

l. 270 – Don't understand "(2) with a relatively poor areal coverage with respect to a standard seismic network" when DAS arrays up to 50-km in length can be deployed.
Covering an area with a continuous fiber cable is much more labour-intensive than covering the same area with a "large N nodes array". Here the area spanned by the DAS cable is smaller than the area covered by the nodes. Conversely, maintaining the cable could be much cheaper. We will clarify these concepts.

l. 271-272 - The statement about "old style" neglects recent deployment of large N node (e.g., Long Beach). So the proper comparison is between large N nodes and DAS.

l. 276 – How does scattering of late arriving surface waves affect P? In general, the wavelengths of P mean that segment-scale heterogeneity or topography should not be cause poor wavefield coherence. Variable coupling of the cable seems more likely.

l. 283-285 – Are "not-aligned segments" the same or different than "three separated segments." i.e., is the difference 2 vs. 3, or parallel vs. not parallel? What is the pointì here?

l. 289-290 - Maybe add that teleseismic waves coming into low-velocity, near-surface sediments are nearly vertical and horizontal DAS sensitivity is attenuated by its cos^2 dependence. Also, emphasize that this is a consequence of the PoroTomo geology and is not necessarily a generalization about DAS.

Thanks for the suggestion; these points will be added in the revised MS.

l. 302 - Clarify meaning of "intra-profile channels". Statement seems contradictory. How can very coherent channels be dominated by locally-scattered waves, unless the profile length is small compared with scatterers?

Pertinent observation. By intra-profile channels we mean a group of adjacent, closely-spaced channels whose spatial coverage is comparable to the size of a scatterer / velocity heterogeneity. Though exhibiting mutual coherency, those channels would imagine a deformed portion of the wavefront.

l. 311-315. Did not try to estimate magnitude of events, but maybe ML can be applied if there are a large number.

Yes, in principle it may be possible, but the event must be located within the deployment; otherwise it is challenging to obtain reliable locations and hence epicentral distance.

---

## Author Response (AR1)

Dear Editor,

first of all, we wish to thank the three reviewers for their precise and helpful comments. We think that our manuscript has been greatly improved in the present version, discussing the relevant points raised by the reviewer. Please find below our response letter to Reviewer #1, where in black you have the original review, and in blue our responses. Line numbers refer to the new version of the manuscript, if not specified. A reference code is inserted for all relevant reviewers' comments (e.g [R1.1], for the first comment by Reviewer 1) and the same reference code is used In the manuscript with tracked changes, to help following our modifications.

Best regards,
Nicola Piana Agostinetti

**Discussion Paper 'Distributed acoustic sensing as a tool for exploration and monitoring: a proof-of-concept', by Nicola Piana Agostinetti and co-workers.**

**Response to Reviewer #1 (Martijn van den Ende)**

**Reviewer #1**
This study compares a DAS array with a collocated nodal seismometer array in terms of exploration and monitoring performance, described by the authors as (lines 52-54): "For "exploration", we mean the definition of spatial variations of elastic properties in the subsurface at high-resolution. With "monitoring", we have in mind those activities adopted for the real-time detection of "events" (in this case, seismic events)". The authors proceed to detail three procedures for the investigation of the two aforementioned tasks. For the task of exploration, the authors compute relative P-wave arrival times through a cross-correlation method (relative to the mean of the array) for both the DAS and geophone arrays. For the monitoring task, the authors consider one local and one regional earthquake. In the case of the local event, P- and S-arrivals are manually picked from the seismometer data, and DAS P-arrivals are picked automatically. The local earthquake is then located through standard travel time inversion methods. The regional earthquake is analysed by weighted time-domain beamforming of the DAS array.

**[R1]** It is clear that the authors put a lot of effort in these analyses, and they have produced many figures intended to support the claims made in the main text. Unfortunately, not all of these claims are sufficiently well supported by the data, in my opinion. Moreover, the authors do not seem to perform their analyses with the objectives stated in lines 52-54 in mind (no subsurface characterisation, no real-time workflows, no earthquake detection). I would strongly suggest the authors to rethink the focus of this manuscript.

We thank Reviewer #1 for pointing out this flaw in the general overview of the manuscript. We believe that the goals of our study are not clearly explained in the manuscript and they could have been mis-understood (see also Reviewer #2's comments). Moreover, we feel that there is a mis-conception in how we reached these goals. For example, "subsurface characterization" is not our goal (e.g. building a velocity model or recovering subsurface structures), because such task could be done in many different ways and different data

could need different algorithms/approaches. Here we are reporting how the relevant data (in case of "subsurface reconstruction", data are the P-wave arrival times) extracted from DAS recordings compare to the same quantities extracted from "node" recordings. We will make this point crystal clear in the Abstract (lines 2-5-8-13, LL 2-5-8-13, hereinafter) and in the introduction at LL 56-57 and 62-63. We also slightly modify the title, according to Reviewer #2, to focus more on the specific topics we are addressing here (new title is: "Distributed acoustic sensing as a tool for subsurface mapping and seismic event detection and location: a proof-of-concept"). According to the new title, we modify the subsection headings (now they are "Subsurface mapping" and "Seismic event detection and location", instead of "Exploration" and "Monitoring").

Below I will try to explain my concerns, hoping that the authors recognise my criticism as ways for improvement of the manuscript. Some of my comments may have been a result of misunderstanding, as I found a few sections a bit hard to follow; in that case the authors could simply point this out and make some clarifications.
Kind regards,
Martijn van den Ende

Main comments:

**[R1.1] Uncertainties in the relative arrival times**. In Section 3.1, the authors investigate the potential of DAS for exploration, defined as characterising spatial heterogeneities in the phase speed of the medium which manifest themselves in the variations of the relative arrival times. These relative arrival times are estimated through cross-correlation of the waveforms in a 0.5-2 Hz frequency band. The main results of this analysis are presented in Figs. 7 and 8, and the authors discuss trends and deviations therefrom for several DAS segments. When taking Fig. 8 as an example, in which a (non-contiguous) segment of roughly 450m is shown, and assuming an average apparent P-wave speed of 5 km/s (consistent with beamforming analysis), I would expect a maximum difference in relative arrival time of less than 0.09s. However, Fig. 8b shows a variation of up to 0.4s in the DAS array, and about half that in the nodal array. This would suggest to me that the uncertainties in the arrival times are (much) larger than the expected moveout. Potentially the frequency band (up to 2 Hz) is too low for the required precision, or the waveform incoherence introduces additional uncertainties in the arrival time estimation. Regardless, without seeing error bars in Figs. 7 and 8, I am not convinced of any conclusions regarding the comparison between the DAS and nodal arrays, and the potential for DAS for exploration tasks.

To address the request of Reviewer #1, we add uncertainties in the P-wave arrival-times obtained with a Bayesian approach (see Piana Agostinetti and Martini, Sedimentary basins investigation using teleseismic P-wave time delays, Geophysical prospecting, 2019) to make our discussion more focused on the point raised by the Reviewer (method presented at LL 116-119). Regarding the specific point raised by the Reviewer #1 (P-wave time-delay along segment 48), we think that it exactly testify the difficulties found in DAS data analysis. As pointed out by the Reviewer #1, the P-wave time-delay measured from the nodal station (oriented horizontal components,) is really large, with respect to what we could expect from standard estimations. And the recordings along the DAS give the same answer (excluding the portion where we have issues (between X=180 and X=230 m). This is due to the anomalous local propagation of the wavefront as seen from the horizontal recording (which is different from the vertical recording, see the new Figure 8). We added a comment at LL 210-213 about this specific observation. We modified Figures 7 and 8, showing computed

uncertainties, and the results at LL 197-203, including the uncertainties measured with our Bayesian approach. Also, to include the observation of P-wave arrival-times along a cable segment for a completely different frequency band, we repeated the experiment with one of the local event found in the DAS recordings (Appendix 1). The results clearly support our conclusions that P-wave time delays measured on the fiber cable are very similar to those recorded by the geophones (oriented horizontal components, of course), but with a completely different resolution which could help to better understand and interpret local effects.

**[R1.2] Amplitude variations in relation to site effects**. In the last paragraph of Section 3.1, the authors suggest that the observed variations in the maximum P-wave amplitude are associated with the local geology, attesting to the exploration capacities of DAS. When I look at Fig. 10, I see two regions of elevated amplitudes, but interestingly these do not seem to lie on top of the hydrothermal features indicated in yellow. At other locations where the cable crosses various fault strands or is positioned close to the yellow features, I don't see such a pronounced amplification. And if I had to draw a third box in this figure, I would put it to the far left of the map, where the cable is far away from any geological features. So it would seem to me that the correlation between elevated amplitudes and the local geology is fortuitous at best. At the moment I can't quite think of any analysis that could prove more conclusive in this regard, so perhaps it would be best to take out this paragraph and corresponding figure.

Looking at Figure 10, it is clear that the quantity "MaxAmp" reaches consistently (i.e. the same MaxAmp value for more consecutive DAS channels) large values (>0.10) only in the two boxes we overplotted, and there is no other area with such high values. Clearly, such threshold (0.10) is subjective and could be discussed. We agree with the Reviewer that there are other areas where the DAS cable is close to the "warm ground" (yellow areas), without traversing it and without showing such large MaxAmp values. We added a comment about this in the manuscript at LL 235-236. Also, amplitude variations as a function of the frequency content of the signals is investigated (Appendix 2), using the recordings of  three Vibroseis sweep operated from a. Position which is almost aligned with a  cable segment. Our results show that amplitude patterns in the DAS recordings follow those of geophone recordings (again, considering oriented horizontal components), but with a clearly higher resolution, which allows to focus on local effects.

**[R1.3] Objectives of Section 3.1**. When I read the authors' definition of "exploration", I initially expected the authors to perform some kind of subsurface imaging similar to what was performed by the PoroTomo team. If my concerns above regarding the uncertainty in the relative arrival times, and the correlation between the waveform amplitude and local geological features are warranted, then Section 3.1 does not offer much in addition to characterise the "spatial variations of elastic properties in the subsurface at high-resolution" (lines 52-53). Instead of relying on 1 passive source (the Hawthorne earthquake), perhaps the authors could use their automated procedure to analyse the numerous vibroseis sweeps that were performed to obtain robust and systematic anomalies in the relative arrival times, or to separate site amplification from the directional sensitivity of DAS, etc. This will likely be a lot of extra work, but it is something that could strengthen Section 3.1 and warrant its existence if the authors agree with my concerns raised in points 1 and 2 above.
As we clarified at the beginning of this letter, we think there is a bit of mis-understanding about the "exploration" concept and what the Reviewer was expecting. For example, we are

not interested here "to perform some kind of subsurface imaging" because, as we know from some decades of studies in such a field, imaging is 80% the algorithm you use to do it. Here we are interested to see how the relevant data, extracted from DAS recordings, compare to the same data, extracted from "node" recordings. It could be useful to study which imaging algorithm would give the best result with DAS data (given the geometry of the DAS cable compared to the geometry of standard geophone networks, the directionality of the recordings and so on), but this is far beyond the scope of our study. However, we agreed with the reviewer that additional analyses using different seismic sources definitely make the manuscript more robust and we added two appendices (Appendices 1 and 2) where we also analysed the Vibroseis recording along a cable segment.

**[R1.4.1] Local earthquake location uncertainty.** In Section 3.2.1 the authors locate a local earthquake with manually picked phases recorded on the geophones, yielding a hypocentral depth estimate of 450 +/- 40m. Performing the inversion with EDT on DAS yields a hypocentral depth that is practically at the surface. The authors suggest that "this discrepancy likely owes to the lack of observations attributed to S-waves, and the simplified velocity structure adopted for travel-time predictions" (lines 229-230). These hypotheses can be tested by performing EDT inversion of the nodal array using either only the P-wave picks or both the P- and S-wave picks. If nodal EDT inversion with only the P-arrivals also puts the source at the surface, but not with both the P- and S-arrivals, then the lack of S-wave picks is to blame. Otherwise it is likely that there is an intrinsic problem with EDT inversion of DAS data. This would render DAS inadequate for local earthquake monitoring, and so the claim that "DAS recordings can be used for monitoring and exploration purposes and their performance is at least comparable to seismological records if not superior" (lines 321-322) is unwarranted.

Agreed. We performed the test requested by the reviewer (see Table 1). In substance, we found  that: (1) The EDT approach applied to manual P- and S-wave pickings from the nodal array (Test #8) does not yield any substantial modification in hypocentral coordinates, including the depth one; (2) By repeating the EDT inversion using only P-wave readings, obtained either automatically (Test #6) or manually (Test #9), the hypocentral solution gets a bit closer to the deployment, at a shallower depth (236-252 m vs 460 of the reference solution). Thus, it looks like S-wave readings are more relevant than the location algorithm in conditioning the depth estimates. However, a full understanding of the dependence of source location on arrival times data, velocity structure and location algorithm would require a thorough assessment of the topology of the EDT misfit function, a task which is beyond the purpose of the present paper. Moreover: in extending the analysis to other events (located well outside the deployment), we noted that the EDT approach produced an artiicial 'attraction' of sources toward the DAS/Nodal network. As a consequence, we decided to get rid of EDT, and redo the analysis using a conventional travel time inversion.

To conclude, from our analysis we found that in an automated workflow, DAS is comparable to a seismometer array for sources located within or in close proximity of the deployment.

Table 1

| Test # | Deployment | Picking | Method | Wave type | Vp | x | y | z |
|--------|-----------|---------|--------|-----------|-----|---|---|---|
|  |  |  |  |  |  |  |  |  |

| | | | | | | | | |
|---|---|---|---|---|---|---|---|---|
| 1 | NODAL | man | GAU | P,S | 2000 | -0.485937 | -0.692187 | 0.308301 |
| 2 | DASH | auto | EDT | - | 2000 | -0.448437 | -0.717187 | 0.204199 |
| 3 | DASH | auto | GAU | - | 2000 | -0.321875 | -1.99688 | 0.00800781 |
| 4 | DASH | auto | EDT | - | 3000 | -0.476562 | -0.767187 | 0.00400391 |
| 5 | DASH | auto | GAU | - | 3000 | -1.04688 | -1.99688 | 0.00800781 |
| 6 | NODAL | auto | EDT | P | 3000 | -0.489062 | -0.751562 | 0.23623 |
| **7** | **NODAL** | **man** | **GAU** | **P,S** | **3000** | **-0.576562** | **-0.851562** | **0.460449** |
| 8 | NODAL | man | EDT | P,S | 3000 | -0.532812 | -0.798437 | 0.436426 |
| 9 | NODAL | man | EDT | P | 3000 | -0.495312 | -0.732812 | 0.252246 |
| 10 | NODAL | auto | GAU | P | 3000 | -0.61875 | -0.93125 | 0.720703 |

**[R1.4.2]** When comparing the hypocentre locations estimated in the present study to those given by Li & Zhan (2018), there appears to be a (very) large discrepancy. Li & Zhan selected 5 earthquakes from the catalogue of Nathwani et al. (2011) as templates for their template matching study. All of the matched events in the catalogue of Li & Zhan should therefore be closely located to the original 5 epicentres, which are all positioned much farther southwest from the DAS array than shown in Fig. 11 of the present study (placing the inferred epicentres right at the edge of the array).

Rejected. This statement is questionable. In building their catalog of matched events, Li and Zhan (2018) also included events whose correlation with the respective template waveform was very low. As a matter of fact, the specific earthquake analysed in the MS has a peak correlation as low as 0.11 (see Table S1 in the Supplemental Material of Li and Zhan, 2018). As a consequence, the closeness between template and matched hypocenters is not necessarily implied.

**[R1.4.3]** Moreover, the depth of these events was estimated to lie in between 750 and 1250m, and not 450m. So regardless which type of array (geophone or DAS) or inversion method (EDT or absolute) is used, the hypocentre estimates of the authors are very different from the previous estimates.

Agreed. See the reply posted above; the low correlation coefficient of the analysed earthquake does not imply its closeness to the respective template event.
Then: before questioning a discrepancy in hypocentral depths on the order of a few hundred meters between events which are not necessarily co-located, the reviewer should wonder

about the (unknown) location error on the template events of Li and Zhan. Moreover: The estimate of the depth parameter depends markedly on the velocity structure adopted. The 5 template events catalogued by the Northern California Earthuake Data Center, have been located using different network and velocity models. Under such conditions. even for the same event, it would not be surprising to obtain different hypocentral coordinates. Finally: we're not particularly concerned about the absolute location of these events; rather, we're interested in comparing the nodal and DAS performances toward automated source location, under the same processing workflow.

**[R1.4.4]** Normally it would be difficult to tell which study is correct, but fortunately the epicentres estimated by the authors are located right next to the instrumented borehole (well 56A-1, 400m deep) at the southern end of the array. From the borehole data the authors should be able to more precisely estimate the depth of the events, and also the distance if the events are really as close as the authors suggested (the propagation front should be strongly curved).

We may be wrong, but it looks like the vertical DAS cable located within well 56A-1 was not operative during the day of the local earthquake (see DASV metadata at https://gdr.openei.org/files/829/DAS_Borehole_cable_files.txt).

**[R1.4.5]** So before drawing any conclusions regarding the suitability of DAS for monitoring, the authors should first establish a reliable benchmark to compare their results with. Since the DAS inversion procedure is automated (no manual phase picking), I would also like to see the inversion results of multiple local events, instead of just one (which could be a lucky hit).

Agreed. This is a pertinent observation. The revised MS now reports additional earthquakes to better corroborate our conclusions, and to verify the performance of the entire workflow on a larger data set.

**[R1.5.1] Regional earthquake beamforming.** Let me first mention that the manuscript of van den Ende & Ampuero (2020) was recently accepted, and that the version of record is now online. The authors may be interested in reading this revised manuscript, though the revised and newly added sections do not directly pertain to the discussion in the present study. Second, the plane-wave fitting approach (which I think what "PWF" means, but this abbreviation is not defined as far as I can tell) is essentially beamforming in the time domain. In the frequency domain, relative time delays become phase shifts, which form the basis for MUSIC beamforming as used by van den Ende & Ampuero. With MUSIC (and other forms of frequency-domain beamforming) the contribution of each sensor to the overall beampower is weighted by the correlation coefficient similar to Eq. (2) on page 5 of the manuscript. So in essence the PWF method is not very different from frequency-domain beamforming methods. The main difference between this and previous studies would be the stacking procedure that supposedly improves waveform coherence. It would be good to show this directly by comparing the original recordings with the stacked waveforms, and show that the stacking procedure positively affects the coherence/scattering. Having that said, the authors do indicate that they get better results that previously obtained by vdE&A. Looking at Fig. 12, it is a bit hard to verify this claim, since the scatter in panels c and d is quite large. It would be very helpful to include something like a moving average weighted by the correlation coefficient of the azimuth and the apparent velocity to see the best estimate of these quantities, and their confidence intervals (!). If I were to draw a line by eye, I would put the mean azimuth at around 180 degrees and the apparently velocity at 3 km/s, which are very different from 337 degrees and 4-6 km/s mentioned by the authors.

Agreed. Before answering these comments, we wish to clarify a basic aspect. With our re-processing of the Hawthorne earthquake using DAS as a sensor array, it was absolutely not in our intention to question van den Ende & Ampuero (2020) results. Rather, we aimed at

complementing this latter work by deriving time-dependent slowness and coherence estimates using a different approach. Having clarified this aspect, let's proceed with a point-by-point reply to the issues raised by the reviewer.

(1) Yes, the reviewer is right: PWF stands for Plane Wave Fitting. We apologize for not having made that acronym explicit; It has been fixed in the revised MS (line XX).

(2) PWF is substantially different from (either time- or frequency-domain) beam-forming. In PWF, inter-channel time delays are first estimated via cross-correlation, and then used to solve a linear problem in which the kernel matrix is given by the differential array coordinates, and the model parameters are the two (or three, in case of a 3D array) components of the slowness vector. Conversely, beamforming methods are inherently non-linear, and are based on (a) solving the forward problem of predicting time delays (phase shifts) given a slowness vector, and (b) using those prediction to calculate a slant-stack, or beam power, of the time- (phase-) shifted seismograms (spectra). The procedure is iterated using a direct search over the slowness plane, and the solution is given by that slowness for which the power of the beam is maximized. Having said this, it is difficult to tell which technique works better under which conditions. In principle, methods exploiting the properties of the spatial covariance matrix of the signal such as MUSIC are expected to provide superior performances in the case of multiple, interfering signals. On the other hand, the weighting scheme we adopted (eq. 2 in original MS) attributes by far more importance to the well-correlated arrivals.

(3) Effects of stacking. Following the reviewer's suggestion, we carefully checked the effect of stacking. And we have to admit that such an operation does not produce any remarkable improvement to the overall coherence (which, however, is not very surprising in light of the small distance spanned by the DAS channels used for stacking and the wavelengths at play). In the following Figure 1, the 79 stacked DAS traces are compared to the recordings from the central channels of each group of 51 consecutive channels used for the stacking procedure. No obvious differences emerge, and the distribution of the maxima of the cross-correlation function among all the independent channel pairs  is substantially the same. As a consequence, in the revised MS we removed the sentence '*(i) stacking of adjacent channels within individual profiles serves to alleviate losses of waveform coherence due to local medium heterogeneities, thus increasing the overall wavefield coherence throughout the DAS deployment*'

(4) Uncertain ties in azimuth an d velocity estimates. In the revised MS, we recomputed the PWF solutions, also estimating 2-sigma error bounds on azimuth and apparent velocities. In such computation, we assumed a constant uncertainty in the time delay estimate eqqual to twice the sampling interval (i.e., 0.02s).

[Figure]

*Fig. (1) - Top panels: Comparison between stacked and original DAS recordings for the Hawthorne earthquake for a 4-s-long window encompassing the P- (left) and S-wave (left) arrivals. At the bottom, the cumulative probability distribution of the maxima of the correlation coefficients between all independent channel pairs.*

(5) We did not compute any temporal averaging of the slowness solutions. Transient changes in such estimate may in fact produce artifacts.

**[R1.5.2]** Lastly, one of the main objectives of this study is to do "real-time detection of events" (lines 52-54), but Section 3.2 does not address either "real-time" or "detection".
Actually, at pg. 8, lines 250-253, we observe that '*As a matter of fact, the onset of the earthquake signal is marked by an abrupt increase of the average multi channel correlation (Figure 12b) which, in correspondence of the P-wave arrival, peaks to a value which is about 2 times larger than those associated with the preceding background noise*'. Then, at lines 260-261, we conclude that: '....*a simple thresholding on the overall correlation of DAS channels may serve as an efficient operator for detecting the arrival of an earthquake signal*'. However: we recognize that we should have better clarified this point and, so that in the revised MS some additional statements will be reported.

**[R1.6]** Conclusions. The authors conclude their work with 4 statements, of which I believe are not well supported by the data, or are not a result of this study at all.
That's a rather hard statement! Why such an acrimonious treatment ?

I would advise the authors to distil a number of take-away messages from their own results. We'll do our best, promise.

**[R1.6a]** I've already disputed Statement #1 that DAS is equal or even superior to conventional seismometers for monitoring and exploration.

We partially agree on this. By the time of comparing DAS to conventional seismometer deployments, one has also to account for the ease of deployment and maintenance of an equivalent number of channels. Under this perspective, we believe that DAS permits faster and easier installation and maintenance procedures. This point will be clarified in the revised MS.

**[R1.6b]** Statement #2 pertains to the volume of DAS data, but this does not make DAS any different from conventional seismic data in terms of the analyses. Other seismological studies (e.g. Roux et al., 2016, GJI) also deal with very dense arrays, and DAS data can always be subsampled if computational resources are limited. Moreover, data volumes and efficiency have not really been the topic of this study.

The referenced statement is a bit more complex. First, thousands of channels sampled at 1KHz do not really represent a standard in passive seismology practice. Other than computational time and storage issues, such large amount of data also need spatial/ temporal subsampling and, as described in our and several other papers, a selection aimed at removing data which may be considered 'bad' a priori (e.g., channels in proximity of the fiber's corners). But this introduces a further issue, that concern the a-priori discrimination between 'bad' and 'good' data…
So: we recognise that our statement #2 was definitely too synthetic, and we will make an effort to clarify the arguments there reported.

**[R1.6c]** Statement #3 suggests that standard seismological tools cannot be applied to single- component measurements. This is true for polarity analyses, but does not necessarily apply to phase travel time inversions, full-waveform inversion, surface wave tomography, ambient noise interferometry, event detection, beamforming, and possibly numerous other analyses that I forgot to list here. And again, the importance of 1C vs. 3C does not really result from the analyses presented in this study.
We still guess that a DAS deployment may exhibit limitations in the measurement of waves whose polarization vector has poor or null projection along the elongation of the fiber. So: no matter the number of possible applications using DAS data, their effectiveness will always depend on the relationships between the wave direction of propagation and polarization, and the elongation of the sensing fiber. As a matter of fact, inspection of Fig. 11 in the original MS evidences that the automatic phase picker identify as a first break either the P- or S-wave arrivals, depending on the orientation of the fiber with respect to the back azimuth, which in turn condition which of the two different wave types is recorded best.

**[R1.6d]** Statement #4 rehashes previous work on earthquake detection, which was not considered in this study.

This criticism is not very clear; we'll try to better explain statement #4. By the time of evaluating DAS performances for earthquake monitoring, an issue is represented by the fact that it may be difficult for a single fiber to achieve a complete azimuthal coverage of seismogenic regions extended over areas of hundreds of squared kilometers or more. Since

the DAS is inherently a multichannel system, a given seismogenic area could be monitored by multiple DASs, which would thus constitute an *array of arrays.* Under this perspective, a better location procedure would consist in the back-propagation of the wave-vectors evaluated at individual DAS systems.

Technical comments:

**[R1.7.1]** In line 120, the authors set-up a velocity model with Vp = 3 km/s. Feigl et al. estimated an average Vp of 2.1 km/s (across all depths), which seems consistent with the beamforming analysis of van den Ende & Ampuero (very crude estimate, though).

Since we were principally interested in the relative locations between the nodal and DAS deployments, for the sake of simplicity we adopted a homogeneous velocity structure whose velocity was chosen by trial-and-error, i.e. by repeating the locations using a velocity of the half-space in between 2 and 5 km/s, and using the RMS of station residuals as a goodness-of-fit indicator. As for the reference to Feigl et al.: we deem the citation pertains to the work 'PoroTomo Final Technical Report:Poroelastic Tomography by Adjoint Inverse Modeling of Data from Seismology, Geodesy, and Hydrology', with digital identifier https://doi.org/10.2172/1499141. If that's the case, in the tomographic slices shown in Fig. 5.5, we observe P-wave velocity increasing rapidly from ~1000 m/s at the surface to 3000 m/s or more at depths of 200-250 m. So, for a source located at depths of several hundred meters (or even more, as in the case of Li & Zhan template events; see response to point above), the 3000 m/s choice does not appear to be so exotic. However, in the revised MS, we made a step forward, and used a gradient velocity model with P-wave velocities increasing from ~2 km/s at the surface, to 5.3 km/s at 3.5 km beneath the surface.

**[R1.7.2]** This may be one reason why the inverted hypocentres are much closer to the array than previously estimated. Also in this line the authors state that they picked 75 vertical and 48 horizontal geophone channels. Shouldn't the picks be the same for all the channels on a given geophone? Why is there a difference in the number of picks?

Again: Is there any reason for which the selected earthquake should be located close to one of the template events? As for the number of P- and S-wave pickings: In seismological observatory practice, it occurs very often that a given station has a very clear P-wave onset, but much noisier/unclear S-wave arrivals. When, such in our case, a huge number of recordings is available, there's no point in measuring phase arrival times with large uncertainties; as a consequence, the number of S-wave readings may be different from that of P-waves.

**[R1.7.3]** Why did the authors resort to only manual picking for the geophones, and not also do automated picking to assess the quality of the automated picks to compare to the DAS picks?

Agreed. That's a good argument. In the revised manuscript, we conducted automatic picking on nodal data as well, so as to compare the locations from the two deployments under the same conditions (see Subsection 2.2.1 in the revised MS, and the Supplementary Material).

**[R1.8]** I did not quite follow the stacking procedure (lines 123-126). It is mentioned earlier that only 1 channel is used per gauge, so does "11 adjacent channels are stacked, with a 20-channel step" mean that a stack is created over 11 gauge lengths? Are these stacks of delayed waveforms (aligning the first arrival so that it doesn't get smeared out)? Then in line 126 it is mentioned that the array is downsampled to 274 channels, "similar to the nodal array", but for the nodal array only 75 (+ 48?) stations were picked.

The stacking procedure adopted in the 'exploration' part is different from that used in the 'monitoring' one. For this latter case, we performed exactly what is written in the text, by

stacking (for instance) channels 1-11,  21-31, etc…, then assigning to the stacked waveform the coordinates of channels 6 and 26, respectively.

**[R1.9]** Lines 179-180, "Nevertheless, this result supports our workflow where DAS data need to be strictly compared to co-located, re-oriented Nodal data": why is this the case? I don't see how this can be concluded from the preceding paragraph.
Reworded

**[R1.10]** In Figure 8 I would suggest to add breaks in the red/green/blue lines to indicate where each segment starts and ends, and not showing the gaps in between the segments as measured data.
Agreed

**[R1.11]** Lines 234-235: but the automated picks should all be for P-waves and their conversions. Was the picking window so large that the direct S-wave could have been picked by mistake?
This is a pertinent comment. Our automatic picking was performed using a window containing the entire event waveform (1 minute, in this case). The picks were later associated using the DBSCAN clustering algorithm, following the procedure adopted by the ScanLoc module (https://docs.gempa.de/scanloc/current/#confval-clusterSearch.maxSearchDist).
So, yes: the reviewer is right.  From inspection of Figure 11, panel (d) it appears that many pickings are in between the expected P- and S-wave travel times, and just a few picks are consistent with S-wave arrivals. The statement '*Many onsets, therefore, are likely representative of S-wave arrivals, following either direct or scattered paths* ' will be changed to '*Many onsets, therefore, are likely representative of late arrivals from the P-wave coda*'.

**[R1.12]** Section 3.2.1: was the origin time co-inverted in the absolute travel time inversion procedure (for the nodal seismometers)? Or was it fixed by the catalogue origin time? EDT doesn't include the origin time, so this information cannot be used to constrain the hypocentre location, which may be another reason why EDT puts the source at the surface. This can be checked by performing EDT on the nodal array.
In both the 'classic' and EDT location methods, the origin time is calculated *a posteriori* using the travel-times to the maximum-likelihood hypocenter. The tests performed for answering to point #4 above suggest that the lack of S-wave arrivals is more relevant than EDT in putting the source at the surface. However, as already clarified, in the revised MS the EDT method is no longer used

**[R1.13]** To fully comply with the Open Science philosophy of Solid Earth, could the scripts that are used to process the data be made available in a repository (e.g. Zenodo or Figshare), so that others could reproduce the results?
Matlab scripts and data necessary for replicating Figure 12 (i.e., the PWF results) have been uploaded on GitHUB : https://github.com/gilb65/PWF.

Minor comments:

Line 28: "strain" = "particle motion"

Line 43: the URL to Feigl's department page is not very long-term sustainable, and it will probably turn into a dead link within a few years. Better to refer to the GDR repositories.

Line 117: typo in the UTC timestamp

Line 255, tiny detail: van den Ende & Ampuero estimated an apparent P-wave speed of 4-6 km/s (page 919), not 5-6 km/s (although it all falls within the uncertainty).
Corrected in the revised MS

Lines 301-302 and 309-310: vdE&A also performed beamforming on subarrays that spanned the entire DAS array (see their Figs. 10-12).
Thanks for the clarification.

Lines 305-307: this is actually the other way around. By assuming a single plane-wave, any additional "sources" are discarded to the noise space. By minimising the projection onto the noise space, the effect of scattering is minimised if there is a stronger plane-wave with array-scale coherence. The more sources are assumed, the more the results are biased towards scatterers.
That part of the discussion has been removed.

Lines 311-312: see Lior et al. (10.5194/se-2020-219) for magnitude estimation with DAS.
Thanks for the indication; a short mention of this work has been added at the end of Discussion section.

**Response to Reviewer #2**

This paper is a relatively brief focused discussion on methods for how a DAS array can be used to 1) map subsurface heterogeneity ("exploration") and 2) detect and locate seismic events ("monitoring"). The methods are validated and compared with geophone ("node") data co-located with the array in the publicly available PoroTomo data set. The paper achieves its goals and is a useful contribution to understanding how to best to use DAS data. The paper should be particularly valuable to anyone planning a surface DAS array for one or both of the targeted applications. While I recommend publication, I do have a list of editorial comments, suggestions, and questions.

**[R2]** The most significant one is for readability and it is to consider combining the methods and results for the two categories of results ("exploration" and "monitoring"). I do also prefer that "exploration" be called "subsurface site effects" or "subsurface heterogeneity mapping" and the "monitoring" be called "seismic event and location detection". These titles are clumsy but they don't overpromise. After all, the basic result in the "exploration" category is relative site effect time delays of +/- 0.12 seconds.
As discussed in the response letter to Reviewer #1's comments, we think that we have to clarify the manuscript goals and how we reach them. We think that "exploration" and "monitoring" should be more appropriate terms, but we also know that such terms have been used for deeper targets ("exploration") and wider workflows ("monitoring") than what we did present here. We clarify goals and terminology as much as possible in the introduction of the revised manuscript (LL 56-57 and 62-63).

Therefore, I might also suggest a more specific title, e.g., Methods for subsurface mapping and event detection using Distributed Acoustic Sensing: Examples from Brady Hot Springs PoroTomo Experiment

We agreed with the Reviewer #2 and we changed the title to make the manuscript more appealing

Specific Comments, Suggestions, Questions.
l. 1. "PoroTomo" is short for "Poroelastic Tomography" so there is no reason to have the "TOMO" all caps.

Changed

l. 36 - Marra et al. is not DAS and is not relevant to this paper.

Changed

l. 103-104 - No node instrument response corrections were made. Is that relevant?

No.

[R2.2] l. 118-150 – These paragraphs from "Methods" would most seamlessly be followed by the "Location" results (see l. 219).
l. 219 – Note how out-of-context this statement is because its antecedent goes back to l.118-135. It is best to discuss this result right after l. 135. Even though the paper is short, I don't think the reader can be expected to keep this in mind. Consider keeping the "data and methods" and their corresponding results together.
l. 225 – Similarly to above, EDT was referred to back in "methods" section. Also, "Conversely" is a term of logic. The meaning here is "In comparison".

We do not fully agree with the Reviewer #2. There is a logic in first giving all details on methodologies and data and, then, presenting and discussing results. For a more casual reader, the structure is much more attractive, having all emphasis at the end. We opted to specify individual subsections in Data and Methods section, referring to the relevant subsection in the Results section. This fact should help in following the manuscript.

[R2.3] l. 212-216 – Can you relate this result to local site effects presented in Parker et al. or Zeng et al.?

We discuss this point at LL 227-230 and 240-245, to connect our results to previous elastic models. As also requested by Reviewer #3

[R2.4] l. 115-117 – Given over 100 events in the Li and Zhan catalogue, maybe give a liitle more reason for choosing the 14 March 10:41 UTC event. Is it the closest to the array?

Yes! We selected that event since, according to a preliminary inspection of arrival times, it appeared to be located in close proximity to the deployment, in turn exhibiting good visibility in both waveforms and cross-correlations (see Supplementary Materials in Li & Zhan, 2018). Moreover, the Supplementary Materials of the revised MS now report results for additional events, including two of the template events used by Li & Zhan (2018) for which catalog locations are available.

[R2.5] l. 118-150 – These paragraphs from "Methods" would most seamlessly be followed by the "Location" results (see l. 219).
l. 124-125- Is the meaning of "20-channel step" that channels 1-11, 31-41, 61-71, etc. are stacked if Channels 1-100 are a segment?

Following the reviewer's suggestion, the 'Methods' section has been re-organized into subsections, in order to establish tighter links with the corresponding results. As for the

channel stacking: following a comment from Reviewer #1, we slightly modified the stacking procedure, and stacked groups of 11 adjacent channels, with a 30-channels step. That means that, if channels 1-100 are a segment (with corners located at channels 1 and 100), stacking is performed over channels 11-21, 41-51,.... 71-81. In this manner, the final number of virtual channels amounts to 239, very similar to that of the nodal array. Note also that now the number of channels which are removed from corners corresponds to a gauge length (10 m), in response to comment **R2.8**.

l. 179-180 Be explicit that "strictly compared" means that nodal horizontal components need to be vector summed in DAS direction.

**[R2.8]** L.207-209. Can you suggest a reason why the criterion for removing channels needs to be increased to 2 gage lengths from a corner?
There was no particular reason behind this choice, except for a conservative criterion of channel selection. However, in the revised MS we adopted the well-established criterion of removing channels spanning a single gauge length (see the response to comment **R2.5**).

l. 233-234 - "projection" implies vector, not tensor, projection. cos^2 is stronger than cos.

**[R2.14]** l. 250 - "PWF" Plane-wave fitting is never defined.
That's right. The PWF acronym is now properly defined in the 'Methods' section.

**[R2.15]** l. 270 – Don't understand "(2) with a relatively poor areal coverage with respect to a standard seismic network" when DAS arrays up to 50-km in length can be deployed.
Covering an area with a continuous fiber cable is much more labour-intensive than covering the same area with a "large N nodes array". Here the area spanned by the DAS cable is smaller than the area covered by the nodes. Conversely, maintaining the cable could be much cheaper.

l. 271-272 - The statement about "old style" neglects recent deployment of large N node (e.g., Long Beach). So the proper comparison is between large N nodes and DAS.
Inserted
Modified

l. 276 – How does scattering of late arriving surface waves affect P? In general, the wavelengths of P mean that segment-scale heterogeneity or topography should not be cause poor wavefield coherence. Variable coupling of the cable seems more likely.
Partially Agreed. For local earthquakes, P-wave wavelenths can be as short as 100-200m, so that heteroheneities on the scale length of a DAS segment may significantly affectthe propagation. As for the rest, we agree with the reviewer and rephrased the mentioned sentence as 'Local scattering phenomena associated with topographic roughness and / or velocity heterogeneities may negatively interfere with the propagation of the P-wave, potentially overwhelming the first arrivals and biasing the measure of, e.g., relative time-delays.

l. 283-285 – Are "not-aligned segments" the same or different than "three separated segments." i.e., is the difference 2 vs. 3, or parallel vs. not parallel? What is the pointì here?

**[R2.20]** l. 302 - Clarify meaning of "intra-profile channels". Statement seems contradictory. How can very coherent channels be dominated by locally-scattered waves, unless the profile length is small compared with scatterers?

Pertinent observation. By intra-profile channels we mean a group of adjacent channels whose overall spatial coverage is comparable to the size of a scatterer / velocity heterogeneity. Though exhibiting mutual coherency, those channels would sense a locally-deformed portion of the wavefront.

[R2.21] l. 311-315. Did not try to estimate magnitude of events, but maybe ML can be applied if there are a large number.
Yes, in principle it may be possible, and in the revised MS we now mention a study by Lior et al. (2020) which reports promising elements toward magnitude estimates using DAS.

**Response to Reviewer #3**

Earthquake seismology is not my discipline. But I see how this paper pushes forward the comparison of DAS and nodal data at Brady's from the recent Wang and Van Eden papers. However, I would like to see if there are more quantitative ways to incorporate the velocity model of the area instead of always making generalities about local velocity changes. This area has great active seismic studies that should be cited (as suggested in the attached) and included in this paper.

We thank Reviewer #3 for her/his review and the commented PDF. We will compare our results to the local velocity model available, where possible. Our study is more focused on the validation of seismic recordings obtained through the interrogation of a fiber cable. Thus, reconstructing a full velocity model is beyond our scope here.

We here answer to the most relevant comments inserted in the PDF, but we also wish to include/address all the comments/questions found in the PDF.

1. like the other reviewer, I have a hard time with this nomenclature: exploration has been done more with the active seismic studies at Brady's.

   Here, we refer to our answer to Reviewer #1 and #2. Being all comments on the nomenclature consistent, we will revise it according to them.

   **[R3.1]**
2. I believe this could be quantified more using the available velocity models of PoroTomo.
3. Again, I would like to see incorporation of the available velocity models

   We will include a comparison with the 3D local velocity models (e.g. the active seismic tomography) and geology where possible (e.g. LL 227-230 and 240-245)

---

## Author Response (AR2)

Dear Editor,

we wish to thank the Martijn van den Ende for the continuous and fruitful discussion about our results. We agree with him that our results should be more quantitative when we speak about correlation between signal amplification and surface geology, and that the present geological data, available to us, should be integrate with more information. Thus, we decided to follow his suggestion and we modified the abstract and the discussion of such point according to his comments. We also included all the other minors modifications.

Best regards,
Nicola Piana Agostinetti
(on be-half of the co-authors)